# Self-potential signals related to tree transpiration in a Mediterranean climate

Kaiyan Hu[1], Bertille Loiseau[2], Simon D. Carrière[2,4], Nolwenn Lesparre[3], Cédric Champollion[4], Nicolas K. Martin-StPaul[5], Niklas Linde[6], Damien Jougnot[2]

[1]Hubei Subsurface Multiscale Image Key Laboratory, School of Geophysics and Geomatics, China University of Geosciences, Wuhan 430074, China

[2]Sorbonne Université, CNRS, EPHE, UMR 7619 METIS, F-75005 Paris, France

[3]Université de Strasbourg, CNRS, EOST, ENGEES, ITES UMR 7063, 67000 Strasbourg, France

[4]Université de Montpellier, UMR 5243 GM (CNRS/UM/UA), Montpellier, France

[5]INRAE, URFM, Domaine Saint Paul, INRAE Centre de Recherche PACA, Domaine Saint-Paul, France

[6]Institute of Earth Sciences, University of Lausanne, Lausanne, Switzerland

*Correspondence to*: Damien Jougnot (damien.jougnot@upmc.fr)

**Abstract.** Plant transpiration is a crucial process in the water cycle and its quantification is essential for understanding terrestrial ecosystem dynamics. While sap flow measurements offer a direct method for estimating individual tree transpiration,

their effectiveness may be limited by the use of point sensors, species-specific calibration requirements, and baseline uncertainties, particularly the assumption of negligible nighttime flow, which may not always hold. Self-potential (SP), a passive geophysical method, holds potential for constraining transpiration rates, though many questions remain regarding the electrophysiological processes occurring within trees. In this study, we continuously measured tree SP and sap velocity on three tree species for one year in a Mediterranean climate. Using wavelet coherence analysis and variational mode

decomposition, we explored the empirical relationship between tree SP and transpiration. Our analysis revealed strong coherence between SP and sap velocity at diurnal time scales, with coherence weakening and phase shifts increasing on days with higher water supply. We estimated electrokinetic coupling coefficients using a linear regression model between SP and sap velocity variations at the diurnal scale, resulting in values typically found in porous geological media. During dry seasons, the electrokinetic effect emerges as the primary contribution to tree SP, indicating its potential utility in assessing transpiration

rates. Our results emphasize the need for improved electrode configurations and physiochemical modelling to elucidate tree SP in relation to transpiration.

## 1 Introduction

Climate change and anthropogenic activities exert significant influence on terrestrial ecosystems, particularly impacting vegetation (Mottl et al., 2021; Nolan et al., 2018). In anticipation of these growing impacts, understanding the effects

of phenomena such as droughts on forests is essential (Anderegg et al., 2016). Evapotranspiration (ET) stands as a critical component of the water cycle, accounting for approximately 65% of precipitated water at the continental scale (Fisher et al., 2017; Oki and Kanae, 2006). Transpiration, constituting 60–80% of global terrestrial evapotranspiration (Bachofen et al., 2023; Dix and Aubrey, 2021; Jasechko et al., 2013; Kuang et al., 2024; Schlesinger and Jasechko, 2014; Wei et al., 2017), can reach 95% in tropical forests during dry seasons (e.g., Kunert et al., 2017). Monitoring of transpiration is crucial for enhancing our understanding of plant–water relations, ecohydrological processes, and water resource management, with implications for agriculture, ecology, and climate science. Plant transpiration can be measured using various techniques, including sap flow methods that quantify water movement through the xylem (e.g., Goulden et al., 1994; Granier et al., 1996; Kume et al., 2010), porometry to assess leaf-level transpiration and stomatal conductance (e.g., Damour et al., 2010; Zhang et al., 1997), and flux towers (eddy covariance) that estimate stand-scale evapotranspiration, with tree transpiration inferred by partitioning soil evaporation (e.g., Kurpius et al., 2003; Scanlon and Kustas, 2012). The heat-balance method is commonly used to measure sap flow velocity in individual plants due to its accuracy, simplicity, and ability to provide continuous measurements (Poyatos et al., 2016; Smith and Allen, 1996). Heat dissipation techniques are widely adopted because of their ease of implementation and high precision (Granier, 1987; Flo et al., 2019; Poyatos et al., 2021; Wang et al., 2023). However, these methods represent point measurements, reflecting localized flow rather than providing an integrated assessment across the entire stem (Flo et al., 2019; Oliveras and Llorens, 2001). They also require species-specific empirical calibrations and regular "zeroing" recalibration during the night (Dix and Aubrey, 2021; McCulloh et al., 2007; Moreno et al., 2021; Poyatos et al., 2016).

In recent years, geophysical methods have gained considerable attention for studying ecohydrological systems (e.g., Carrière et al., 2021a, b; Dumont and Singha, 2024; Harmon et al., 2021; Hermans et al., 2023; Jayawickreme et al., 2014; Loiseau et al., 2023; Luo et al., 2020; Voytek et al., 2019). Among geophysical methods, the passive self-potential (SP) method has shown promise for estimating sap velocity (Gibert et al., 2006; Gindl et al., 1999). Measurable SP signals naturally occur between trees and the surrounding soil in response to bioelectrical effects within plants. Tree SP measurements are often positive when measuring the voltage difference of the stem with respect to the surrounding soil, the so-called biopotentials (e.g., Fensom, 1957, 1963; Gibert et al., 2006; Tattar and Blanchard, 1976; Zapata et al., 2020). Tree transpiration processes facilitate the transport of water and solutes within the xylem (e.g., Kim et al., 2014). Additionally, sugar concentration gradients can generate turgor pressure differences, driving flow within the phloem (e.g., van Bel, 2003). These natural processes can trigger electrokinetic effects through the advection of net electrical charges and electro-diffusive effects driven by electrochemical potential gradients, leading to the generation of biopotentials and measurable SP signals. In theory, SP signals provide an integrated response between the two electrodes that is sensitive to the transpiration process.

SP signals are often measured by non-polarizable electrodes and a high-impedance voltmeter to avoid current leakage arising from the measurements. Due to their stability, non-polarizing Ag/AgCl electrodes have been used to measure SP signals within the trunk in laboratory experiments (e.g., Gil and Vargas, 2023; Gindl et al., 1999; Oyarce and Gurovich, 2010). However, this kind of electrodes contains gels or electrolytes that dry up after one to two months for small-sized electrodes,

thereby, leading to unreliable measurements (Fensom, 1963). Such electrodes are hence deemed impractical for prolonged, continuously monitored field experiments. Hubbard et al. (2011) developed a miniaturized Petiau-type Pb/PbCl$_2$ electrode (Petiau, 2000) for measuring SP induced by a biogeobattery within a small laboratory column, potentially offering a suitable solution with an extended lifespan. However, its installation presents challenges, particularly in shallow depths beneath bark and to ensure effective contact with the sapwood. Instead, in outdoor natural environment, polarizable stainless metal electrodes are often used for tree SP measurements (e.g., Fensom, 1963; Gibert et al., 2006; Le Mouël et al., 2024; Zapata et al., 2020, 2021). A non-polarizing electrode is one in which the electrode potential is independent of the current passing through the circuit. However, in both polarizable and non-polarizing electrodes, electrode-related effects are superimposed on the primary effects of interest, often leading to observations that are difficult to interpret. The electrode potentials depend on the electrode design, the temperature, as well as the surrounding and internal hydrochemical environments (Jougnot and Linde, 2013). Due to electrode-related effects, the measured electrical voltage shows often important drifts during long-term monitoring, related for instance to electrode aging (e.g., Perrier et al., 1997; Hu et al., 2020). Hao et al. (2015) conducted experiments to measure plant-related voltages using electrodes made of different metals and found differences exceeding 10 mV when using copper electrodes with 5 mm differences in diameter. When employing metal electrodes to measure SP on trees, it is hence crucial to ensure consistency in terms of electrode design (materials and geometry). Consequently, the installation and the choice of electrodes used to measure tree SP should be made very carefully.

Even in the absence of electrode effects, the interpretation of tree-SP signals is complicated due to the different mechanisms contributing to the observed data. These include both electrokinetic and electrochemical processes occurring within the sapwood (Fensom, 1957; Fromm and Lautner, 2007, 2012; Gibert et al., 2006; Gil and Vargas, 2023; Love et al., 2008; Tattar and Blanchard, 1976). Moreover, Barlow (2012) and Le Mouël et al. (2024) suggest that luni-solar tides could induce electric fields, possibly further complicating the interpretation of SP measurements on trees. However, some long-term measurements have not observed clear monthly patterns related to lunar cycles (e.g., Fensom, 1963, Zapata et al., 2021). Uncertainties surrounding these mechanisms make quantitative interpretations of SP data in terms of the water dynamic within the soil-plant system very challenging.

Given the considerable ambiguity surrounding the mechanisms governing the coupling magnitude, which determines the mathematical relationship between SP and sap flow observations, further experimentation is essential to clarify their correlation in situ. The existing literature on plant-related SP predominantly focuses on individual trees (Belashev, 2024; Gibert et al., 2006; Hao et al., 2015; Le Mouël et al., 2010; Voytek et al., 2019) or different tree species, often with non-continuous or short-term measurements (e.g., Gil and Vargas, 2023; Oyarce and Gurovich, 2010; Pozdnyakov, 2013; Zapata et al., 2020). Recently, Le Mouël et al. (2024) analysed SP signals from various trees using the singular spectrum decomposition method, suggesting that sap flow and multi-scale tree SP may be partially or entirely induced by terrestrial gravitational tides, although without providing data on sap flow. Several pertinent questions remain unanswered regarding tree SP observations: (1) Do different tree species exhibit distinct time-varying SP patterns under identical environmental conditions? (2) Do trees of the same species display SP characteristics that depend on the weather conditions? (3) Will two

trees of the same species show similar SP responses to sap velocity? (4) Is SP a reliable indicator of sap velocity within the trunk? (5) When are electrokinetic effects dominating tree SP responses?

To address these questions, we review tree SP theory in Section 2, followed by a description of long-term SP monitoring experiments conducted at three distinct sites in the French Mediterranean region, along with an outline of data analysis methods in Section 3. In Section 4.1, we present and analyse the data obtained from measurements on four different trees at three sites. Sections 4.2 and 4.3 investigate the short-term characteristics of tree SP in terms of correlations with sap velocity, precipitation, evapotranspiration, and other meteorological data. Finally, Section 5 discusses these results by quantitatively relating tree SP signals to various electrical potential sources such as hydraulic properties and electrochemical effects.

## 2 Theoretical background concerning tree SP

Pioneering research of tree SP monitoring dates back to the early 1960s, when Fensom (1963) embarked on extensive studies to monitor SP on different trees. Earlier studies of the bioelectrical potential can be found in Fensom (1957). Fensom (1963) observed that the electrical voltages were the strongest during growing seasons and proposed that voltage measurements of plants might detect cambial growth. Subsequent studies by Gindl et al. (1999) measured both sap velocity and SP on tree stems in a well-controlled laboratory experiment, uncovering a significant linear correlation between the two. They suggested that the transpiration process acts as a driving force in xylem sap, generating electrokinetic effects that explain the relationship between the sap velocity and SP data, suggesting the potential value of using SP data to estimate plant sap flow. Through continuous observations of an individual poplar tree at various heights and positions, Gibert et al. (2006) found that the relationship between the SP and the sap-flow velocity is not entirely linear. They suggested the presence of electrically active structures within the xylem leading to electro-diffusive effects within plants. Voytek et al. (2019) simulated the SP generated by root water uptake and reconstructed the observed time-varying patterns of SP in the root zone, confirming that plant root water uptake affects the observed electrical voltages. However, the numerical simulations accounting for the electrokinetic coupling process induced by the soil water movement in the root zone could not predict the amplitude of the observed SP.

Previous experimental studies on SP between the living plant and the surrounding soil have shown that the electrical voltages generated by physiological, physical, and chemical processes can range from tens to hundreds of millivolts (e.g., Fensom, 1963; Pozdnyakov, 2013; Zapata et al., 2020). These magnitudes are comparatively high compared to SP signals generated by near-surface water or solute flux (e.g., Jougnot et al., 2015; Voytek et al., 2019; Hu et al., 2020). A schematic diagram of a tree is presented in Fig.1 to summarize the mechanisms of SP generation. SP differences from root to shoot are induced within the sapwood, including the xylem and phloem (Figs. 1b&c). Below, we describe the possible mechanisms generating tree SP.

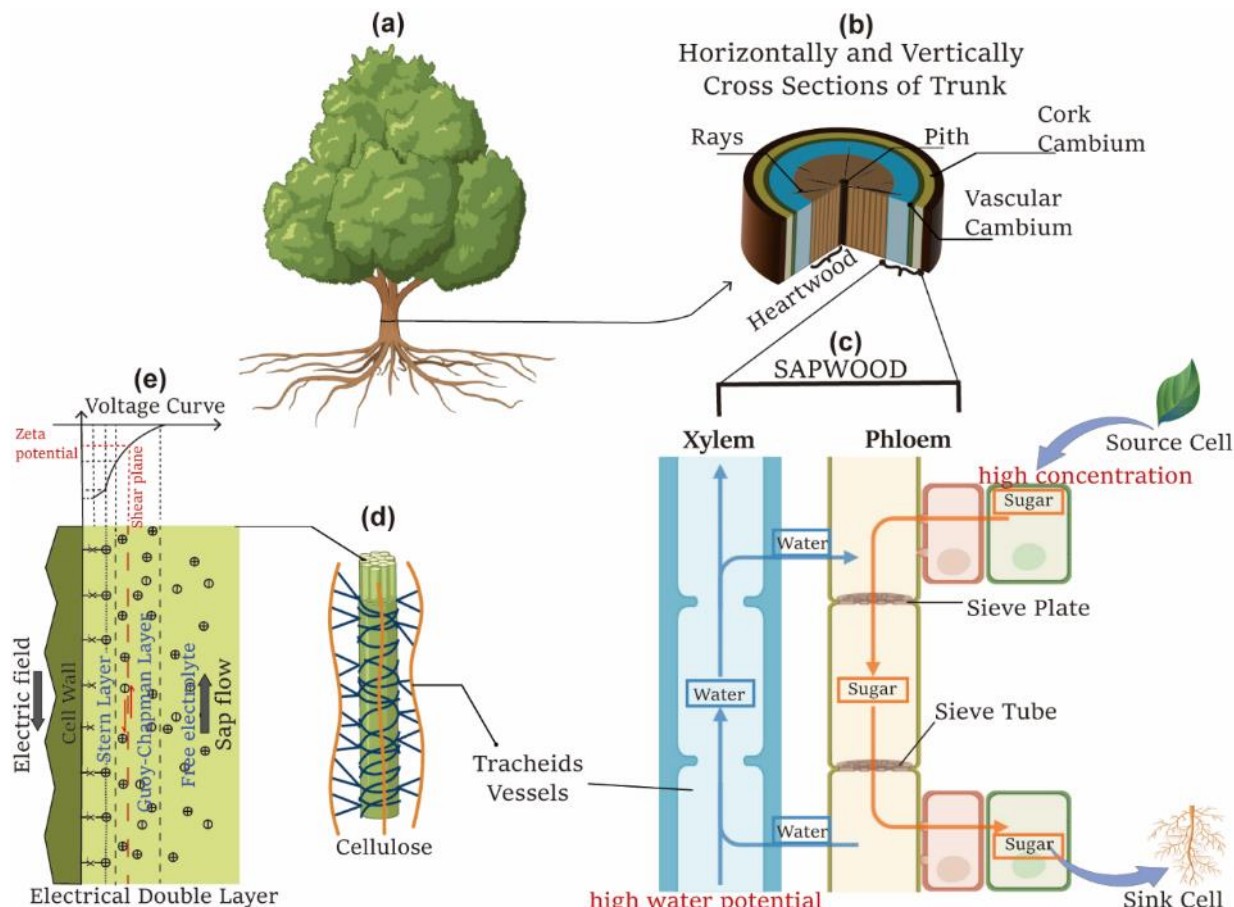

**Figure 1: Schematic diagram of tree SP: (a) Adapted tree illustration from "Ash tree" by BioRender.com (2024); (b) Horizontally and vertically cross sections of trunk; (c) Vertically extracted xylem and phloem adapted from "Plant Vessels Layout" by BioRender.com (2024); (d) Cellulose structure adapted from "Cell wall (cellulose)" by BioRender.com (2024); (e) Electrical double layer.**

### 2.1 Self-potential signals generated by xylem and phloem transport

Observed SP signals arise from the superposition of contributions arising from different coupling sources and from electrode-related effects (e.g., Hu et al., 2020; Jougnot et al., 2015; Jougnot and Linde, 2013). The SP signal due to flow and mass transport process can be described by the following Poisson equation in absence of external electrical current:

$$\nabla \cdot (-\sigma \mathbf{E}) = \nabla \cdot (\mathbf{J}_{ek} + \mathbf{J}_{diff}), \tag{1}$$

where $\sigma$ (S·m$^{-1}$) is the bulk electrical conductivity; $\mathbf{E} = -\nabla V$ (V·m$^{-1}$) is the electrical field; $\nabla V$ is the electrical potential gradient; $\mathbf{J}_{ek}$ (A·m$^{-2}$) and $\mathbf{J}_{diff}$ (A·m$^{-2}$) are the electrokinetic and electro-diffusive current density, respectively. We detail these two contributions below.

The primary contribution of sap ascent occurs in the xylem, where vessels transport water and dissolved minerals from the roots throughout the plant. This process relies on transpiration pulling, creating a negative pressure gradient that

drives upward flow through the xylem (Fig. 1c). The xylem vessels feature a capillary-like structure, as illustrated in Fig. 1d. Although the cellulose walls of the xylem are electrically neutral and insulating, they become negatively charged as they attract

water molecules (Fensom, 1957). For negatively charged cell walls, the streaming current, generated by the upward movement of excess charge within the Gouy–Chapman diffuse layer as sap flows, represents the advective transport of electrical charges, resulting in a net source of current density $\mathbf{J}_{ek}$. According to the electrokinetic mechanism in presence of an electrical double layer, the upward flow of sap induces a natural electric field in the opposite direction, as depicted in Fig. 1e.

The electrokinetic cross-coupling coefficient $C_{ek}$ (V·Pa$^{-1}$) can be estimated by the ratio of voltage difference $V_{z1} -$

$V_{z2}$ to pressure difference $p_{z1} - p_{z2}$ with no external currents (i.e., electrical flux equals 0). This describes a condition where only electrokinetic effects are present, with no contribution from diffusive currents in Eq. (1), ensuring that the net current remains zero. The $C_{ek}$ is determined using the Helmholtz–Smoluchowski (HS) equation:

$$C_{ek} = \frac{V_{z1}-V_{z2}}{p_{z1}-p_{z2}}\bigg|_{\mathbf{J=0}} = \frac{\epsilon_f\zeta}{\sigma_f\eta_f}, \tag{2}$$

where $\epsilon_f$ (F·m$^{-1}$) and $\sigma_f$ (S·m$^{-1}$) denote the dielectric permittivity and the electrical conductivity of the capillary flowing fluid, respectively. The parameter $\zeta$ (V) is the Zeta potential at the shear plane of the electrical double layer (Fig. 1e); it typically has

155 a negative sign. Both $\sigma_f$ and $\zeta$ are influenced by temperature and ion concentration. Consequently, the voltage induced by xylem transport not only depends on sap velocity but also on sap temperature and concentration. Based on laboratory experiments under different pore water ionic concentrations, Linde et al., (2007) proposed an empirical relationship between the logarithm of $C_{ek}$ and $\sigma_f$ for porous media. A similar model could be devised to estimate $C_{ek}$ within vascular plants.

Alternatively, the electrokinetic current density $\mathbf{J}_{ek}$ (A·m$^{-2}$) can be expressed by the product of the effective excess

charge density $\hat{Q}_v$ (C·m$^{-3}$) and the Darcy velocity $\mathbf{u}$ (m·s$^{-1}$) (e.g., Jougnot et al., 2020):

$$\mathbf{J}_{ek} = \hat{Q}_v\mathbf{u}. \tag{3}$$

Here, $\hat{Q}_v$ indicates the density of excess charge in the diffuse layer of electrical double layer model (Fig. 1e) that is effectively dragged by sap flow. Following Kormiltsev et al. (1998) and Jougnot et al. (2019), $C_{ek}$ can be transformed into $\hat{Q}_v$ by

$$\hat{Q}_v = -\frac{\eta_f\sigma}{k} C_{ek}, \tag{4}$$

where $k$ (m$^2$) denotes permeability of a porous medium. Under the assumption of 1–D flow and neglecting $\mathbf{J}_{diff}$ in Eq. (1), the current density is solely governed by the electrokinetic effect. The equivalent current density induced by the sap flow can be

simplified to:

$$\sigma\mathbf{E} = -\sigma\nabla V = -\hat{Q}_v\mathbf{u}. \tag{5}$$

Thus, the $\hat{Q}_v$ can be expressed as (e.g., Jougnot et al., 2012)

$$\hat{Q}_v = \frac{\sigma\nabla V}{\mathbf{u}}. \tag{6}$$

During the transpiration process, water loss coupled with osmotic changes lead to an increase in solute concentration towards the crown. This implies the potential generation of electro-diffusive potentials due to electrochemical differences. The xylem sap typically contains approximately 10 mM of inorganic nutrients, along with organic nitrogen compounds metabolically synthesized in the root (Nobel, 2009). Common ion species found in plants include protons ($H^+$), calcium ($Ca^{2+}$), potassium ($K^+$), magnesium ($Mg^{2+}$), ammonium ($NH_4^+$), chlorine ($Cl^-$), nitrate ($NO_3^-$), among others (Davies, 2006; Miller and Wells, 2006; Volkov and Markin, 2012). For instance, increases in $K^+$ of xylem sap occur with increasing transpiration demand and consequently increase hydraulic conductivity and capacity of water movement (e.g., Losso et al., 2023; Nardini et al., 2011). However, the concentration distribution along the longitudinal direction exhibits different trends and patterns among different trees (McDonald et al., 2002). According to McDonald et al. (2002), for a Norway spruce tree in late autumn, the total concentration of amino acids and the minerals magnesium, calcium, and potassium in the xylem sap ranges from 4 mM at the ground level to 6 mM near the crown base. This concentration range aligns with typical salinity levels in pore water (Linde et al., 2007). The electro-diffusive current density $\mathbf{J}_{\text{diff}}$ ($A\cdot m^{-2}$) arises from the electrochemical potential gradient, influenced by the activities of different ion species:

$$\mathbf{J}_{\text{diff}} = -\frac{k_B T_K}{e_0}\sigma \sum_{i=1}^{N} \frac{t_i^H}{q_i}\nabla \ln a_i, \tag{7}$$

where $a_i$, $t_i^H$ and $q_i$ represent the activity (i.e. thermodynamically effective concentration), the microscopic Hittorf transport number, and the valence of the ion species $i$, respectively; $k_B$ ($K\cdot J^{-1}$), $e_0$ (C), $T_K$ (K), and $\sigma$ ($S\cdot m^{-1}$) are the Boltzmann constant, the elementary charge, the temperature and bulk electrical conductivity; $N$ is the total number of ion species in xylem sap. The equivalent electrical potentials induced by $\mathbf{J}_{\text{diff}}$ is expressed by

$$\nabla \varphi_{\text{diff}} = \frac{\mathbf{J}_{\text{diff}}}{\sigma} = -\frac{k_B T_K}{e_0}\sum_{i=1}^{N}\frac{t_i^H}{q_i}\nabla \ln a_i. \tag{8}$$

Compared with the electro-diffusive effect, cell membrane potentials across xylem tracheids constitute another source of electrical potential gradient (Fensom, 1957; Hedrich and Schroeder, 1989). Further details on the membrane potential can be found in Nobel (2009) and Spanswick (2006).

Water movement within the xylem is mainly sustained by hydrostatic pressure gradients driven by transpiration processes. Conversely, in the phloem, water flow is influenced not only by hydrostatic pressure gradients but also by osmotic pressures resulting from translocation processes. Specifically, the loading of sugars into the phloem leads to an increase in solute concentration but a decrease in water potential in source cells, creating a higher water potential in sink cells (Fig. 1c). Consequently, water and solutes move upward in the xylem and downward in the phloem (Hölttä et al., 2006). Moreover, osmotic pressures in the phloem cause water from the xylem to radially flow into the phloem (Fig. 1c). Apart from the passive electrokinetic and electro-diffusive mechanisms, actively transporting charged particles into cells may lead to the accumulation of internal electrical potential and the occurrence of electrogenic pumps across membranes (Nobel, 2009).

## 2.2 Previous studies and discussion of tree SP generation

Multiple studies have explored the relationship between sap flow and electrical potential differences along plant stems, often attributing it to electrokinetic effects (e.g., Gindl et al., 1999; Gil and Vargas, 2023; Gibert et al., 2006; Fensom, 1963; Koppán et al., 2002). However, this explanation has come under scrutiny due to inconsistencies observed in measurements. Love et al. (2008) conducted a laboratory experiment involving *Ficus benjamina* trees and their surrounding soil for different pH levels. They proposed that disparities in electrical potential differences between the plant and its surrounding soil are predominantly influenced by the pH difference between the xylem and the soil. Similarly, Zapata et al. (2020), based on measurements from various trees in a Mediterranean forest region, suggested to abandon the electrokinetic mechanism to explain SP differences along the trunk. However, their measurements were conducted over a limited period. Furthermore, fluctuations in pH alter the concentrations of $H^+$ and $OH^-$ ions in aqueous solutions, consequently affecting the Zeta potential of the electrical double layer model (e.g., Al Mahrouqi et al., 2017; Leroy et al., 2013). As a result, pH variations contribute to changes in the electrokinetic coupling coefficients and the magnitudes of corresponding SP.

Aside from the observed longitudinal differences in electrical potential, measurements taken across the stem's cross-sectional profile reveal distinct radial polarization (Isam et al., 2017; Zapata et al., 2020). Isam et al. (2017) conducted measurements of short-circuit currents radially along the trunk's growth direction, noting a peak current near the vascular cambium—a layer between the secondary xylem and phloem (refer to Fig. 1b). They observed strong electrical current generation may result from the low electrical resistance of living tissues, the influx of water into the phloem, and the diffusion of charges around the cambium region. These findings suggest that the measured tree SP varies also with respect to the depth at which electrodes are inserted into the trunk.

As the measured SP is influenced by multiple mechanisms, its composition changes with water and solute movements, exhibiting spatial and temporal variations. Interpreting tree SP observations becomes complex due to these multifaceted mechanisms, which exhibit different patterns under various seasonal and meteorological conditions. Rather than relying on a single generative mechanism and short-term observations, we present below results obtained from continuous, long-term monitoring of tree SP, sap velocity, and meteorological data to comprehensively analyse their correlations over time.

## 3 Materials and Methods

### 3.1 Study sites and trees

Three distinct experimental sites were selected in the French Mediterranean region to monitor sap velocity and SP on four trees from three different species (Fig. 2). The characteristics of each site are summarized in Table 1. All sites are located into karstic hydro-systems. The Larzac and LSBB test sites are part of the H+ national observatory service (hplus.ore.fr) and the OZCAR research infrastructure (ozcar-ri.org; Gaillardet et al. 2018), while the Font-Blanche site belongs the ICOS research infrastructure (Gielen et al., 2017).

The Font-Blanche Forest, located 30 km east of Marseille and approximately 7 km from the coastline (Fig. 2a), sits at an elevation of 425 m (Ollivier et al., 2021). The bedrock consists of sub-recifal Cretaceous limestone with Urgonian facies. The forest is dominated by Aleppo pines (*Pinus halepensis Mill.*), averaging 13 m in height, with an understory of Holm oaks (*Quercus ilex* L.) with an average height of 6 m (Girard et al., 2012; Simioni et al., 2020). Electromagnetic induction and electrical resistivity tomography (ERT) characterized the spatial variability as a function of soil depth (Carrière et al., 2021a).

**Table 1** Features of experimental sites and trees

| Property \ Sites | Larzac | LSBB | Font-Blanche | |
|---|---|---|---|---|
| Tree number | C2 | A3 | P327 | C325 |
| Acronyms | LaQp | LSQi | FBPh | FBQi |
| Tree species | Pubescent oak (*Quercus pubescens*) | Holm oak (*Quercus ilex* L.) | Aleppo pine (*Pinus halepensis Mill.*) | Holm oak (*Quercus ilex* L.) |
| Circumference of trunk (m) | 0.900 | 0.428 | 1.083 | 0.396 |
| Xylem anatomy | Ring-porous | Ring-porous | Diffuse-porous | Ring-porous |
| Electrical resistivity of sapwood ($\Omega \cdot m$) | 282 | / | 369 | 250 |
| Bed rock | Dolomite | Limestone | Limestone | |
| Mean altitude (m) | 708 | 422 | 425 | |
| Soil depth (cm) | 0–80 | 0–70 | 0–60 | |
| Mean annual precipitation (mm) | 870 | 900 | 650 | |
| Mean annual actual evapotranspiration (mm) | 600 | 580 | 460 | |
| Mean annual temperature (ºC) | 10.5 | 13 | 14 | |

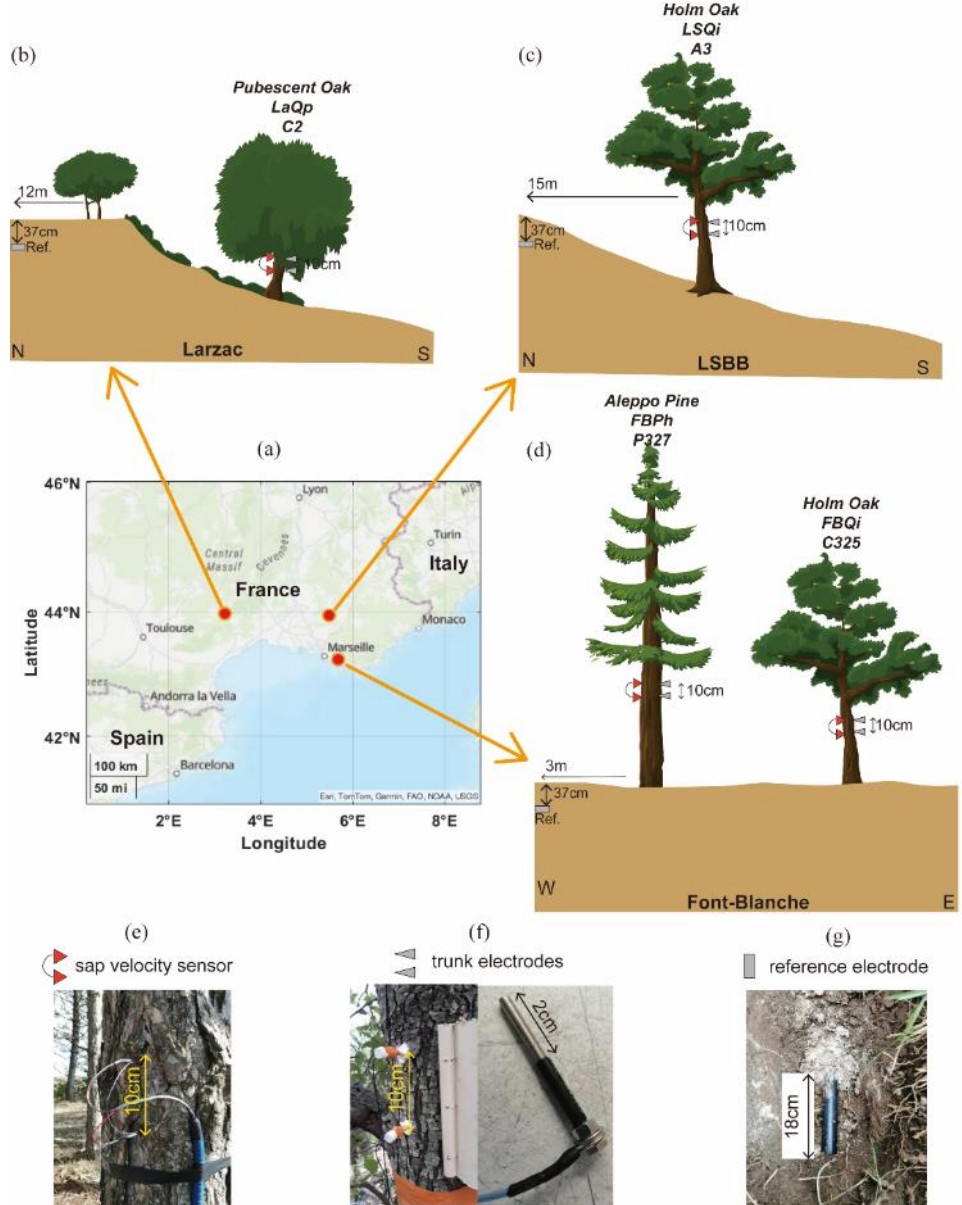

**Figure 2: The observation system at the three forest sites: (a) Location map of sites indicated by red circles; (b–d) Observation setups at the Larzac site, the LSBB site, and the Font-Blanche site, respectively; (e–g) Sensors for measuring sap velocity, electrodes for measuring tree SP, reference electrodes buried underground.**

The LSBB site, located in the southern part of the Fontaine-de-Vaucluse karst hydrosystem at an altitude of 422 m, is dominated by Holm oak (*Quercus ilex* L.), averaging 13 m in height. The bedrock composition is similar to Font-Blanche. Detailed studies of the site's structure and hydrodynamics were conducted by Carrière et al. (2016) using ERT, ground-penetrating radar, and magnetic resonance sounding. Carrière et al. (2020a, b) examined interindividual drought responses

along a transect monitored by ERT. Evapotranspiration at the mountain scale was studied using superconducting gravimetry (Carrière et al., 2021b).

The Larzac observatory, situated at an average altitude of 708 m, features a landscape of grasslands, black pine plantations, and deciduous stands dominated by Pubescent oaks (*Quercus pubescens*), with monitoring focused solely on the oaks. The observatory is part of the Durzon karstic system, primarily composed of dolomite, and has been explored using

gravimetry (Fores et al., 2018; Jacob et al., 2008), ERT, and seismic methods (Valois et al., 2016).

Four trees have been equipped with SP and sap velocity sensors (Fig. 2 and Table 1): 1) *Quercus pubescens* at the Larzac observatory with a trunk circumference of 90 cm, 2) *Quercus ilex* L. at the LSBB site with a circumference of 43 cm, and 3) *Pinus halpensis Mill* and *Quercus ilex* L. with circumferences of 108.3 cm and 39.6 cm, respectively, at the Font-Blanche site. These measurements allow us to compare their physiological characteristics of different species under different

meteorological and geological conditions.

### 3.2 Instrument and data

### 3.2.1 Sap velocity measurements

Data were collected at the three sites throughout 2023 (Table S1 of Supplementary Material). The raw data related to this study can be found in Hu et al. (2024). To assess transpiration rates, sap velocity was measured using the heat dissipation

technique by Granier (1987) at the four trees. Two thermocouple probes were inserted 2 cm into the sapwood of each tree, positioned 10 cm apart. The upper probe applied a constant heating power (see Fig. 2e). The temperature difference between the upper and lower probes is influenced by sap flow surrounding the heated probe. The Granier-type heat-dissipation probes were installed at around 1.5 m above the ground of the trees. Thus, by continuously heating in the upper probe, sap velocity could be estimated by measuring the temperature difference between the probe pairs by copper constantan thermocouples

under the thermal imbalance. The original signal was recorded in millivolts, subsequently transformed into temperature with a calibration coefficient (~0.04 mV·$^o$C$^{-1}$, Do and Rocheteau, 2002). Then, the sap velocity was calculated by the Granier's empirical equation (Granier, 1987) with the obtained temperature differences. After calibration and correction, the sap velocity is obtained in $\mu m \cdot s^{-1}$ at 30-minute intervals (e.g., Moreno et al., 2021).

### 3.2.2 SP measurements

The electrical voltages were measured using a Campbell Scientific CR1000X datalogger using the voltmeter function with an input impedance of 20 GΩ. The sampling time intervals for tree SP measurements are 1 minute at the Larzac and LSBB sites, and 10 minutes at the Font-Blanche site. Details about the measurement parameters, sampling intervals, and units can be found in Table S1. Each tree is equipped with a pair of stainless-steel screws with 3 mm diameter, separated vertically by 10 cm, serving as trunk electrodes (Fig. 2f). These electrodes were horizontally aligned with the corresponding sap flow

sensors on each tree (Fig. 2). After peeling away the bark from the trunk, the electrodes were inserted to a depth of 2 cm to

ensure contact with the sapwood. The outside part of the electrodes was coated with black insulating rubber (as depicted in Fig. 2f). All trunk electrodes were identical in material, size, and shape. Following installation, the exposed portions of the electrodes were further shielded with epoxy glue to isolate them from environmental factors and minimize the impact of meteorological conditions such as air temperature, humidity and radiation.

Polarizable stainless-steel electrodes were used in previous outdoor experiments for tree SP measurements, such as Gibert et al. (2006) and Zapata et al. (2021). Gibert et al. (2006) discussed the temperature effects of steel electrodes and found that artificially heating and cooling a tree electrode did not result in significant variations in electrical potential compared to another electrode that was not thermally disturbed. In this work, we neglect temperature effects on electrode potentials. Non-polarizable Petiau-type (Pb/PbCl$_2$) electrodes (Petiau, 2000) were selected as reference electrodes and buried in the soil at a depth of 37 cm (Fig. 2g). The raw signal is a voltage between the trunk and the reference, which includes the electrical potential of tree in situ and electrode. In this study, we used the voltage measured at the lower electrode minus the higher electrode to indicate the tree SP.

### 3.2.3 Meteorological data

The physiological conditions of trees respond to climate changes. We gathered data on actual evapotranspiration (Actual ET), precipitation (Prec.), vapor pressure deficit (VPD), and air temperature (Temp.) from the meteorological station located at the Font-Blanche site. Additionally, a meteorological station near to the LSBB site supplied us with daily precipitation, mean air temperature, and global radiation data. Hourly air temperature and precipitation data were obtained from a meteorological station situated less than 10 km from the Larzac site. Detailed information refers to Table S1 of the Supplementary Material.

### 3.3 Data analysis

#### 3.3.1 Wavelet analysis of sap velocity and tree SP

To identify correlations and phase lags between sap velocity and tree SP at different time scales, we calculated the coherences between the two time-series using continuous wavelet transformation coefficients. The wavelet coherence $C_{\text{SV,SP}}(a, b)$ between sap velocity (SV) and SP signals is calculated by:

$$C_{\text{SV,SP}}(a,\ b) = \frac{|S(W_{\text{SV}}(a,b)W_{\text{SP}}^{*}(a,b))|^2}{S(|W_{\text{SV}}(a,b)|^2)S(|W_{\text{SP}}(a,b)|^2)}, \tag{9}$$

where the superscript * denotes the complex conjugate; $S$ denotes the smoothing operator function (Torrence and Compo, 1998); $W_{\text{SV}}(a,b)$ and $W_{\text{SP}}(a,b)$ are the continuous wavelet transforms of sap velocity and SP signal at the wavelet scale $a$ and position (frequency or period) $b$ using the Morlet wavelet (e.g., Grinsted et al., 2004; Linde et al., 2011).

#### 3.3.2 Variational mode decomposition

Due to factors such as temperature variations, trunk wounds, and electrode polarization effects, the recorded SP data

may exhibit long-period drifts. Additionally, SP measurements are susceptible to high-frequency electromagnetic noise interference. For example, Fourier spectrum analysis of SV data for FBPh and FBQi through 2023 indicates the presence of one-, two-, three-, and four-day$^{-1}$ frequency signals (Figs. S1a, b in the Supplementary Material). However, the SP spectrum (Figs. S1c, d) does not exhibit clear dominant frequencies that can be directly characterized through Fourier analysis. As introduced in Section 2.1, the generation mechanisms of tree SP are complex and multiple. This means that the measured SP in our experimental sites contain signal contributions originating from various sources. Similar to an electrocardiogram (ECG), the tree SP is an electrophysiological signal. The variational mode decomposition (VMD) approach has been applied to process ECG signals and extract the intrinsic modes from signals to obtain information-containing spikes and low-frequency patterns (long-term drifts) related to physiological activities.

VMD can be used to decompose signal $f(t)$ into sub-signals $u_k(t)$ with different centre frequencies $\omega_k$ (Dragomiretskiy and Zosso, 2014):

$$f(t) = \sum_{k=1}^{N} u_k(t), \tag{10}$$

where $k$ and $N$ denote the number and total number of decomposed modes. Assuming each mode $u_k$ to be concentrated around a centre frequency $\omega_k$ in its bandwidth, the below minimization problem is used to determine the decomposition $u_k$:

$$\min_{\{\omega_k, u_k\}} \left\{ \sum_{k=1}^{N} \left\| \partial_t \left[ \left( \delta(t) + \frac{j}{\pi t} \right) * u_k(t) \right] e^{-j\omega_k t} \right\|_2^2 \right\}, \tag{11}$$

where $j^2 = -1$, $\delta(t)$ and $*$ denote the Dirac function and convolution, respectively. More details about the optimization process to solve Eq. (11) with updating the modes and the centre frequencies can be found in Dragomiretskiy and Zosso (2014). In this study, we use the frequencies with the centre of gravity and the largest amplitudes of the mode's power spectrum to represent the centre frequency and the dominant frequency of each corresponding mode, respectively.

## 4 Results

### 4.1 Overview of one year of raw data

#### 4.1.1 Raw time-series data

The tree SP data collected at the Font-Blanche site are displayed in Fig. 3 (refer to Section 3.2.2). The time-varying sap velocities and SP obtained on the Aleppo pine trunk (Figs. 3c&e) differ from those of the Holm oak trunk (Figs. 3d&f), even when the two trees are in proximity and experience identical weather conditions (Figs. 2d, 3a&b). Both trees show rather large SP signals with a more pronounced response to environmental changes on Aleppo pine compared to Holm oak. However, the overall time-varying patterns of the raw SP data from trees do not demonstrate a clear correlation with sap velocity, as evidenced by relatively low Pearson correlation coefficients (see Fig. S2 in Supplementary Material). Additional data from the Larzac and LSBB sites are provided in Fig. S3.

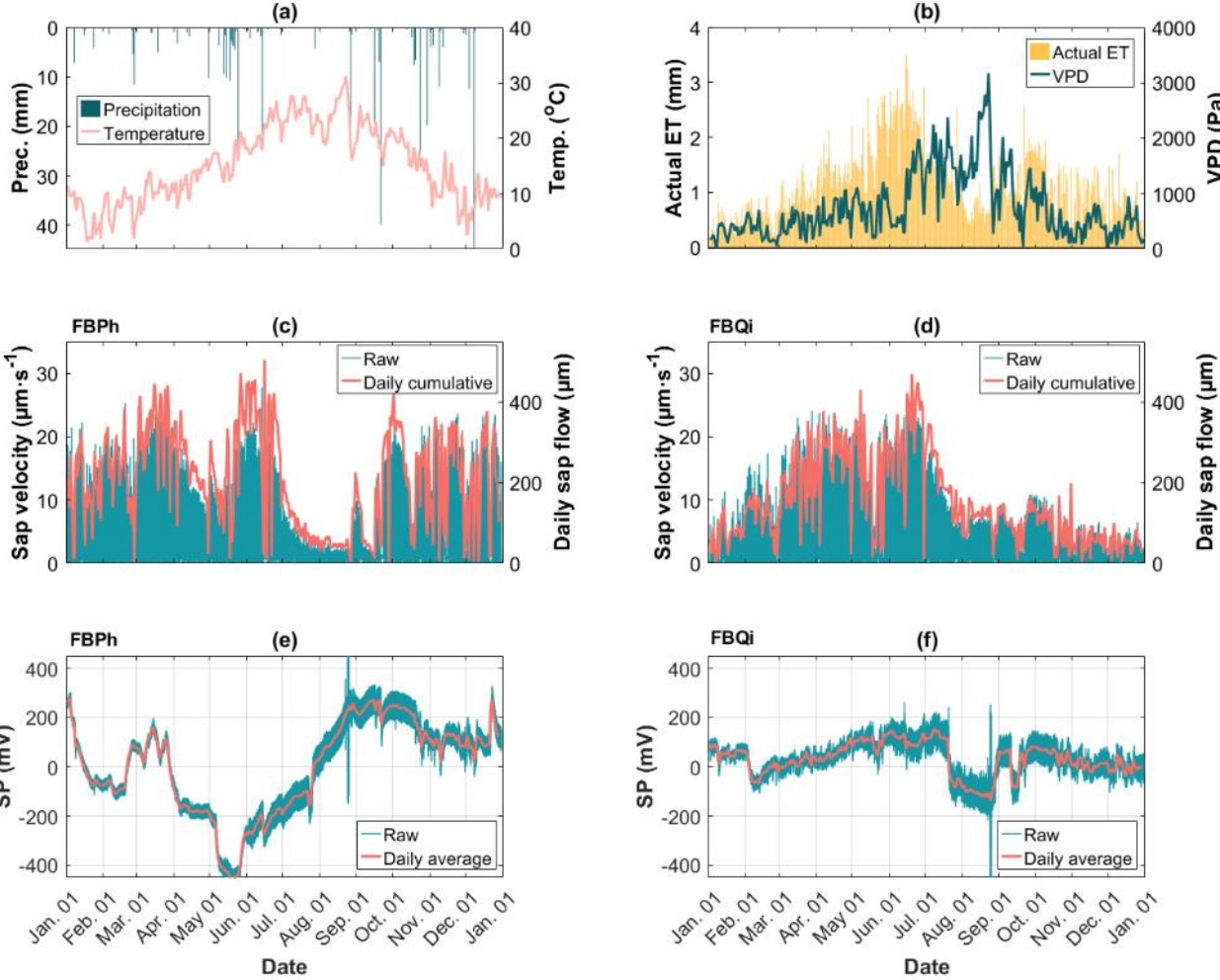

**Figure 3: One-year data collected at the Font-Blanche site at half-hourly intervals from January 1, 2023, to January 1, 2024. (a)**
330 **Precipitation and air temperature data; (b) Actual evapotranspiration (Actual ET) and vapor pressure deficit (VPD) data; (c–d)**
**Measured sap velocity in blue lines and daily cumulative sap flow in pink lines for the Aleppo pine (FBPh) and the Holm oak (FBQi),**
**respectively; (e–f) Measured SP data in blue lines and daily average of SP data in pink lines for the Aleppo pine (FBPh) and the**
**Holm oak (FBQi), respectively.**

### 4.1.2 Wavelet analysis of sap velocity and tree SP

According to Section 3.3.1, we calculated the wavelet coherence between the sap velocity and tree SP data collected
throughout 2023. As an example, we present the coherence maps for the two trees at the Font-Blanche site in Fig. 4. Wavelet
coherence indicates a localized correlation in time-frequency space (e.g., Grinsted et al., 2004). Figures 4b–c illustrate the

temporal correlation between SV and SP across different time periods. The arrows in Figs. 4b–c represent the phase relationship between SV and SP. Particularly, the rightward arrows (0° phase) indicate in-phase behaviours, where SV and SP vary synchronously. Leftward arrows (180° phase) indicate anti-phase behaviours, signifying negative correlation or out-of-phase variations between SV and SP. Conversely, upward or downward arrows (±90° phase) reflect phase shifts, suggesting that one signal leads or lags the other by a quarter cycle. In particular, the one-day period shows high coherence, with arrows pointing left, indicating that SP is negatively correlated with SV at this timescale. This indicates that SP and SV exhibit stronger correlation at a daily timescale compared to other periods. Notably, tree SP exhibits a negative correlation with sap velocity across the two different species of trees. Note also that the coherence diminishes during periods of water surplus (Fig. 4a). In contrast, the Holm oak at the LSBB site shows lower coherence, and the Pubescent oak at the Larzac site lacks coherence between sap velocity and SP (Fig. S4). The wide discrepancies for different trees at different test sites might be attributed to different annual rainfall (Tabel 1 and Figs. S3a–b) and other factors such as electrochemical and electrode-related effects contributing to the measured SP data.

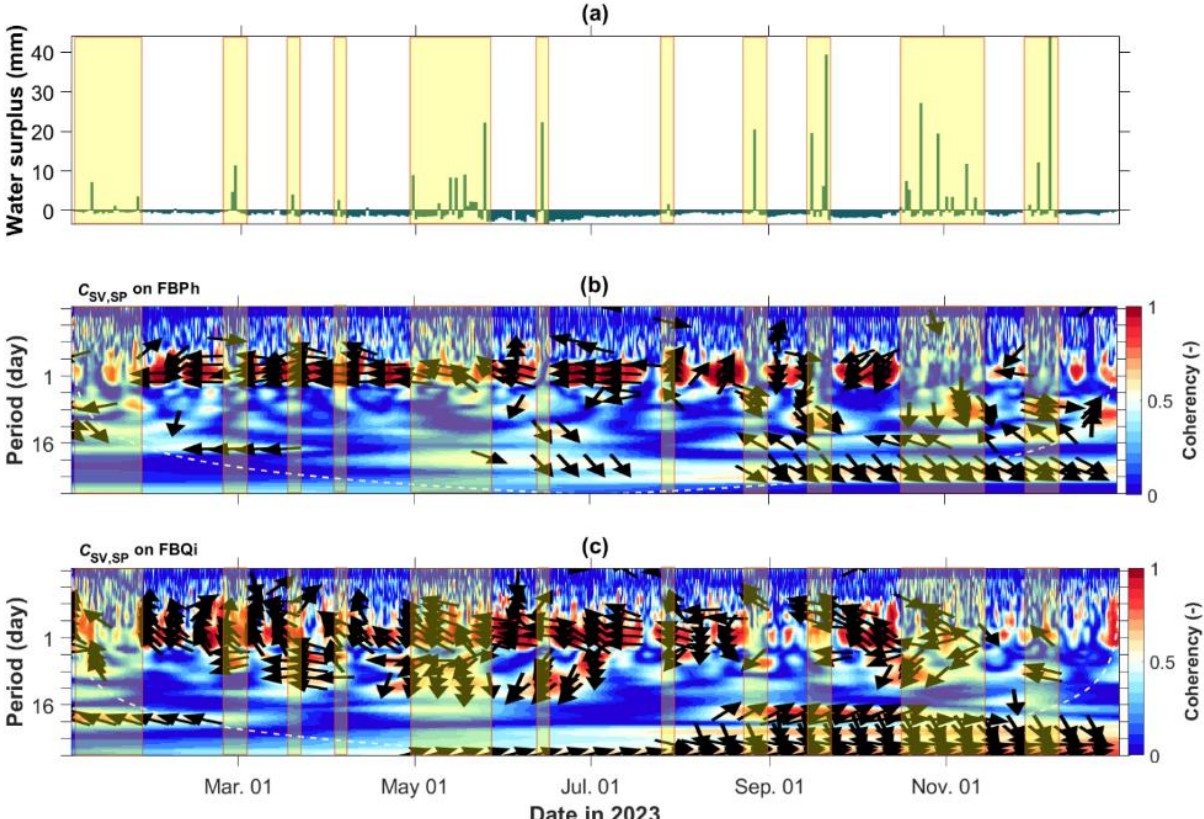

Figure 4: **Wavelet coherence analysis between sap velocity and tree SP data at the Font-Blanche site in 2023. (a) Water surplus calculated as the difference between precipitation and actual evapotranspiration. Wavelet coherence maps for the (b) Aleppo pine and (c) Holm oak at the Font-Blanche site with yellow highlighted boxes indicating periods of water surplus. Arrows denote the lag/lead phase between the two timeseries; white dashed lines (b-c) indicate the cone of influence within which edge artifacts are negligible.**

## 4.2 Decomposition results

In our case, VMD is used to separate the measured data into a finite set of components (modes). We initially assume six principal modes as all coefficients of determination ($R^2$) between the measured tree SP, sap velocity data with the sum of the corresponding six intrinsic modes using one-year data from 2023 for all four trees are 1.00, meaning that the results of VMD could effectively reconstruct the measurements. As an example, we present the frequency spectra of six modes decomposed from the tree SP and sap velocity (SV) obtained from the Holm oak tree at the Font-Blanche site throughout 2023 (Fig. 5). The centre and dominant frequencies of each decomposed mode decrease sequentially. Besides the expected diurnal variations driven by the rhythm of sap flow (Figs. 5i–j), other high-frequency content and low-frequency trends are considered noise and long-term drifts in this study.

When compared to the decomposition spectrum of SV data (Figs. 5b&d), noise present in the tree SP data tends to manifest at higher frequencies, as indicated by the prominent spectral amplitudes in the higher-frequency range (Figs. 5a, c, e, g). The low-frequency components, which reflect baseline trends, dominate the last intrinsic modes and occupy the largest proportion of the signals, as shown by the highest amplitudes in the SP and SV spectra (Figs. 5k–l). Apart from the fifth intrinsic mode, the decompositions of SP and SV exhibit distinct frequency characteristics, highlighting the complexity of SP signals. Notably, the fifth intrinsic modes for both datasets reveal a strong diurnal rhythm, with a dominant frequency of 1 d$^{-1}$ (Figs. 5i–j). The spectra of the fifth decomposition also display the second-highest amplitudes, indicating that these diurnal components represent a significant portion of the overall SP and SV signals.

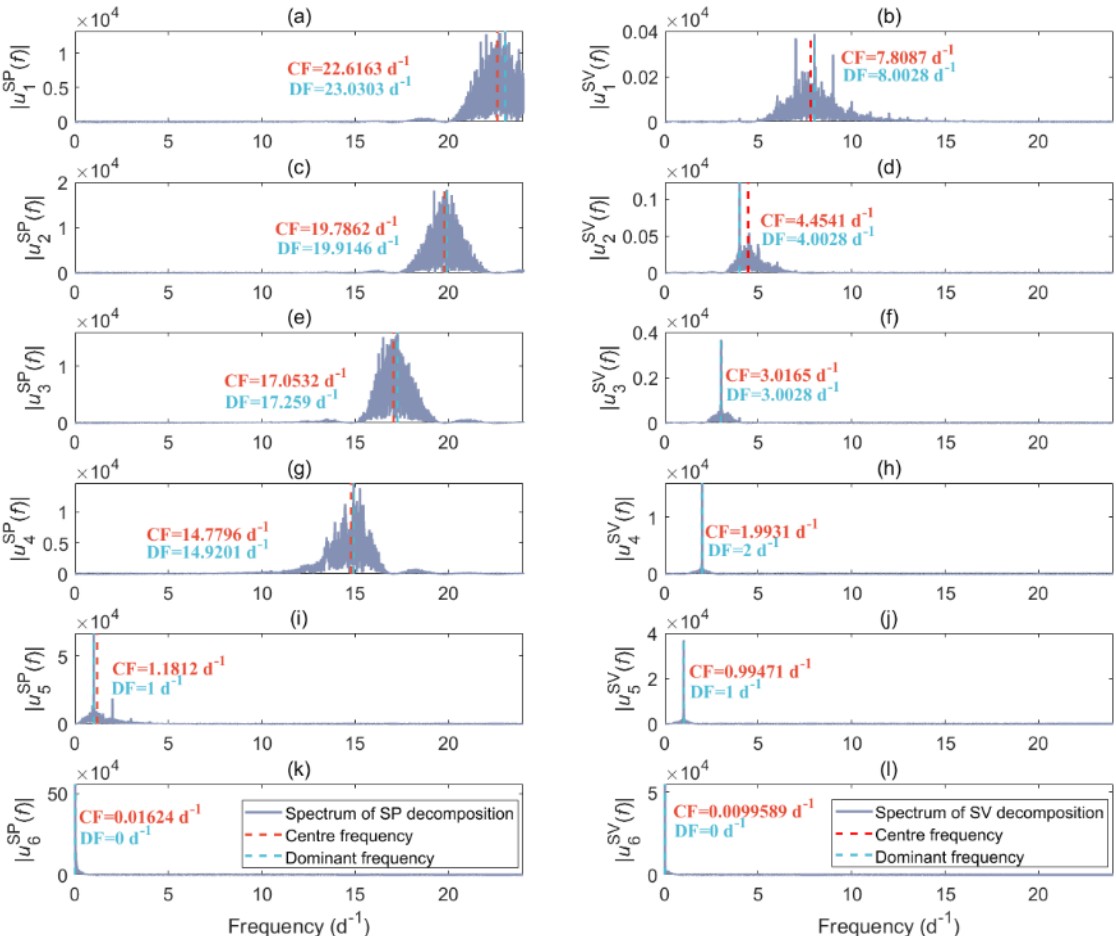

**Figure 5: Frequency spectra of six decomposed modes of (left column: a, c, e, g, i, k) tree SP and (right column: b, d, f, h, j, l) sap velocity data obtained on the Holm oak (FBQi) at the Font Blanche site within 2023 using the variational method (VMD); Different rows correspond to different modes, where "CF" and "DF" indicate the centre frequency and dominant frequency of the corresponding mode, respectively.**

We investigated the impact of increasing the number of modes from 6 to 12 when performing data decomposition. Regardless of the number of modes used, the results consistently indicate a diurnal rhythm in the second last intrinsic mode with similar amplitudes and patterns (refer to Fig. S5). We also conducted an analysis on the tree SP and sap velocity data collected from the neighbouring Aleppo pine (FBPh) using the VMD approach (Fig. *S6*). The frequency bands of each decomposition derived from FBPh are comparable to those from FBQi. Particularly, the high-frequency oscillating characteristics of SP components on both trees are nearly identical. Since our primary interest lies in understanding the diurnal variations associated with transpiration, we employed the same VMD procedure to the whole datasets to extract the fifth decomposed results based on decompositions with six modes.

### 4.3 The relationship between SP and physiological characteristics of trees

#### 4.3.1 Correlation coefficients between data collected at the Font-Blanche site

Tree transpiration is affected by water availability, VPD, net radiation, and atmospheric uptake of water (e.g., Alfieri et al., 2018). Employing the VMD approach and extracting six intrinsic mode functions, we delineated the diurnal rhythms using the fifth subset component of tree SP, sap velocity and meteorological data collected during a two-week period in the growing season of 2023. The two-week period from April 16 to April 30 was selected to capture a phase of active transpiration across different species, during which no precipitation was recorded at the Font-Blanche site (as shown in Fig. 3). The large long-term fluctuations in raw SP data across positive and negative ranges, as well as high-frequency interferences, are excluded by extracting the fifth decomposed mode. This allows us to focus on relative variations aligned with the dominant sap flow frequency, providing clearer insight into the relationship between SP and SV.

The fifth VMD components exhibited significant correlations between tree SP and sap velocity, notably at the Aleppo pine (FBPh) with a correlation coefficient of -0.88. The cross-correlation coefficients between decomposed sap velocity and SP for the Holm oak (FBQi) trees reached -0.66. Based on the electrokinetic mechanism generated by sap flow in Section 2.1, a larger negative SP should indicate a higher transpiration rate, which is indicated by the negative signs of the Pearson correlation between the sap velocity and tree SP in this case. Interestingly, the fifth extracted VMD components of tree SP data on FBPh displayed relatively high correlations with other data, including tree SP at FBQi (see Fig. S7). Below, we further analyse the correlations between the time-varying tree SP with the sap velocity data on different trees and sites.

#### 4.3.2 Relationship between tree SP and sap velocity across four trees

To analyse the linear relationship between sap velocity and SP, scatter plots are displayed in Fig. 6 using the fifth VMD modes. The fifth decomposed modes of SP and SV signals are denoted by "$VMD_5^{SP}$" and "$VMD_5^{SV}$", respectively. The fifth VMD results of tree SP and SV data present similar frequency content (Figs. 5i&j). Here, we assume that the fifth VMD results of tree SP originated from diurnal variations of sap velocity. Please note that the negative values in the fifth decomposed mode of SV signals represent relative deviations from the baseline, reflecting diurnal fluctuations rather than indicating actual reverse sap flow. The baseline component is captured in the sixth decomposed mode, isolating physical sap flow from the oscillatory variations seen in the fifth decomposed mode. Upon applying linear regression of SP versus SV, for two species of trees investigated at the Font-Blanche Forest, the Holm oak (Fig. 6c) exhibits a steeper slope compared to the Aleppo pine (Fig. 6a). Outliers were removed under a 95% confidence level by the linear regression to obtain the slope coefficients. In contrast, the correlation of decompositions between the tree SP and SV is lower on the Holm oak at the LSBB (Fig. 6e) than on the same species at the Font-Blanche (Fig. 6c). Additionally, there is no correlation found on the Pubescent oak at the Larzac site (Fig. 6g).

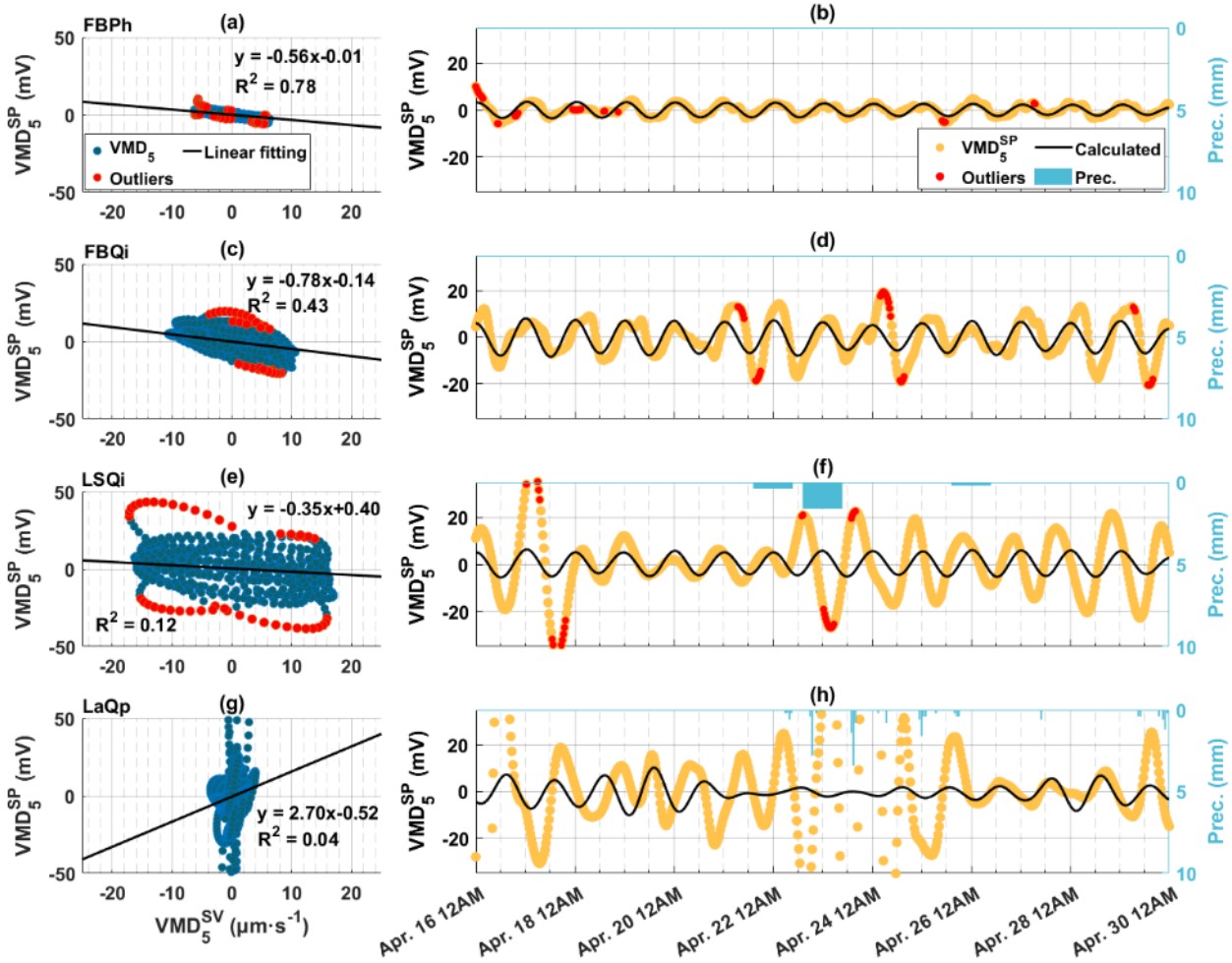

**Figure 6: The relationship between the fifth decomposed modes of sap velocity ($VMD_5^{SV}$) and tree SP ($VMD_5^{SP}$) signals in April 16–30, 2023, at the Font-Blanche, LSBB, and Larzac site. (a, c, e, f) Scatter plots of the fifth decomposed modes of sap velocity and SP signals for (a) the Aleppo pine (FBPh), (c) the Holm oak (FBQi) at the Font-Blanche site, (e) the Holm oak at the LSBB (LSQi), and (g) the Pubescent oak at the Larzac (LaQp), with black lines indicating linear regression results and red dots indicating points (outliers) outside the 95% confidence level of the regression. (b, d, f, h) Corresponding decompositions of SP signals (orange dots) and calculated SP (black lines) based on the linear relationship between $VMD_5^{SV}$ and $VMD_5^{SP}$, with red dots as outliers outside the 95% confidence level of the linear regression.**

Using the parameters obtained from the linear regression (Figs. 6a, c, e, g), we predicted the tree SP responses as a function of the sap velocity (refer to Figs. 6b, d, f, h). The predictions effectively replicate the time-varying SP patterns observed in the Aleppo pine at the Font-Blanche (Fig. 6b). Rainfall events decreased the coherence and increased the phase shifts between sap velocity and tree SP (Fig. 4). In this case, due to the rainfall events during the extracted period at both LSBB

and Larzac sites, the $VMD_5^{SV}$ could not restore the $VMD_5^{SP}$ obtained from the measured data (Figs. 6f&h). Additionally, the modelled tree SP for the Holm oak at the Font-Blanche fails to accurately capture the highly varying amplitudes observed in the measured SP data (Fig. 6d). Specifically, elevated SP amplitudes are noted on April 21, 22 and 24.

Since the amplitude of the calculated SP (derived from the fifth decomposed mode of SV signals, $VMD_5^{SV}$) is lower than that of the fifth decomposed mode of SP signals ($VMD_5^{SP}$) for Holm oak at the Font-Blanche site on April 21–22, and 24 (Fig. 6d), we conducted a test to investigate whether the VMD method reduces the relative amplitude of SV signals during decomposition. To achieve this, we compared the raw SP and SV data to the sum of the diurnal features ($VMD_5^{SP}, VMD_5^{SV}$) and the baseline component ($VMD_6^{SP}, VMD_6^{SV}$) obtained from the VMD-processed data (Figs. 7a–b). The combination of the fifth and sixth decomposed modes is analogous to low-pass filtering, retaining lower-frequency signals. The highest diurnal amplitudes of the raw and processed SV data remain relatively stable, whereas SP data shows notable variations, particularly on April 21–22 and 24.

To further investigate whether the underestimation of SP using $VMD_5^{SV}$ resulted from the exclusion of high-frequency components, we examined the sum of the first to fifth decomposed modes of SV signals, representing a detrending process. In Fig. 7c, the fifth decomposed mode of SP ($VMD_5^{SP}$) was compared with the SP calculated from $\sum_{i=1}^{5} VMD_i^{SV}$. This comparison demonstrated that even when all decomposed modes except the baseline were used to reconstruct diurnal SP from SV data, the higher SP amplitudes observed on April 21–22 and 24 could not be reproduced (Fig. 7c). To corroborate this observation, we repeated the analysis using a finite impulse response (FIR) filter, which yielded similar results (Fig. 7d). This suggests that the elevated SP magnitudes observed on these dates may be influenced by factors other than sap flow.

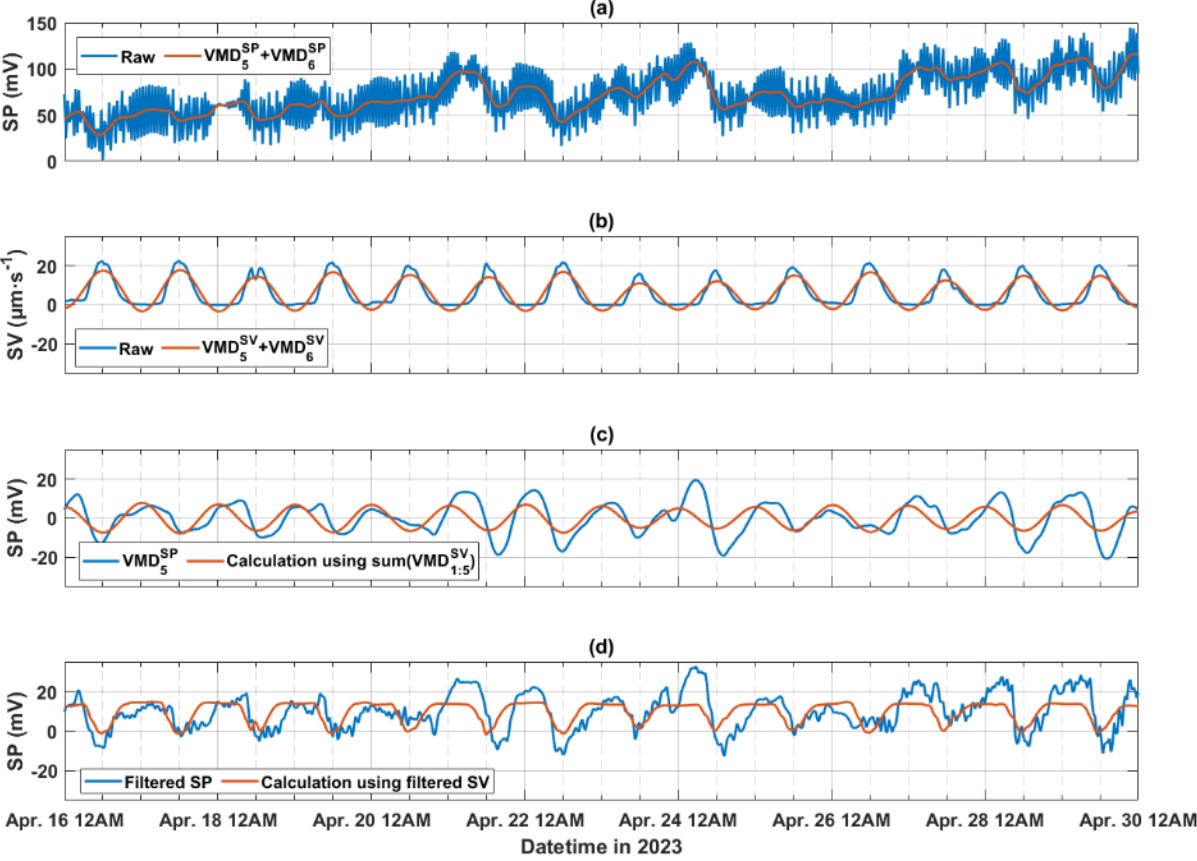

**Figure 7: Two-week SP data during the 2023 growing season for Holm oak (FBQi) at the Font-Blanche site processed using different approaches. (a) Raw SP data compared with the sum of the last two decomposed models of SP signals; (b) Raw sap velocity data compared with the sum of the last two decomposed models of SV signals; (c) Fifth decomposed mode of SP signals, alongside SP calculated from the sum of the first to fifth decomposed modes of SV signals ($\sum_i^5 VMD_i^{SV}$); (d) SP data filtered using a finite impulse response (FIR) filter (bandpass: 1/48 h⁻¹ to 1/1.5 h⁻¹; filter order: 100) compared with SP calculated from FIR-filtered SV data using the same filter parameters.**

As detailed in Section 3, the height of the lower electrode with respect to the upper electrode on the monitored trunks is -10 cm, allowing us to calculate the electrical potential gradient ($\nabla V$) between the measurement points using $VMD_5^{SP}$. Moreover, electrical resistivity tomography has been conducted on trunks at both the Font-Blanche and Larzac sites. The average sapwood resistivities of the Aleppo Pine and the Holm oak at the Font-Blanche are 369 $\Omega\cdot$m and 250 $\Omega\cdot$m, respectively (e.g., Moreno et al., 2021). By converting the localized $\nabla V \approx -\frac{VMD_5^{SP}}{10\,[cm]}$ and $\mathbf{v} \approx VMD_5^{SV}$ to the International System of Units, we estimated the effective excess charge density ($\hat{Q}_v = \frac{\sigma \nabla V}{\mathbf{v}}\Big|_{J=0}$) using linear fitting slope coefficients ($\frac{\nabla V}{\mathbf{v}}$). Due to the lack of

resistivity measurements at LSBB (LSQi) and the presence of rainfall at the Larzac and LSBB sites during the selected periods, excess charge density calculations were limited to the two trees at the Font-Blanche site, where sap flow consistently dominated SP generation. The resulting $\hat{Q}_v$ values for FBPh and FBQi in spring are 15.2 C·m$^{-3}$ and 31.2 C·m$^{-3}$, respectively. Calculating the seasonal variations of $\hat{Q}_v$ yields ranges of $6.8 - 25.7$ C·m$^{-3}$ for FBPh and $31.2 - 68.0$ C·m$^{-3}$ for FBQi (Fig. S8 and Table S2). These values are consistent with those typically found in porous media (pH ranging from 6 to 8.5), such as sandstones, glass beads, and limestones (Jardani et al., 2007; Revil and Jardani, 2013). These materials correspond to typical pore sizes of $10 - 100$ µm, which is of the same order of magnitude as the diameter of the cells conducting the sap (e.g., Sperry, 2003).

## 5 Discussion

### 5.1 Hydraulic properties and tree SP

Water flow along the soil–plant continuum is governed by hydraulic conductivities and water potential gradients. While the water movement in sapwood differs from those in the vadose zone, certain functional similarities exist. For instance, the flow in xylem and soil can both be approximated by Poiseuille's Law (e.g., Jougnot et al., 2019; Nobel, 2009) and exhibit similar responses of hydraulic conductivity to water potential gradients (Cai et al., 2022). Although xylem hydraulic conductivity does not fluctuate with minor variations in soil water potential, significant reductions in soil moisture can lead to declines in xylem conductivity, particularly under drought conditions (Brodribb and Hill, 2000; Carminati and Javaux, 2020). When soil water potential falls within a moderate range, xylem hydraulic conductivity remains relatively stable. However, at lower soil water potentials, xylem conductivity decreases (Cai et al., 2022; Kröber et al., 2014).

For the Font-Blanche site, the selected two-week periods shown in Figs. 8 and S9 correspond to different seasonal water dynamics (Fig. S10). As indicated in Fig. S10b, soil water potential is lower in spring, and summer compared to winter and autumn. Notably, the slopes (ratios) of $\text{VMD}_S^{\text{SP}}$ to $\text{VMD}_S^{\text{SV}}$ peak during summer, with similar but lower values in winter and autumn. An empirical relationship between the hydraulic permeability $k$ and $\hat{Q}_v$ for porous materials is given by Jardani et al. (2007):

$$\log_{10}(\hat{Q}_v) = -9.23 - 0.82\log_{10}k. \tag{12}$$

This relationship is also reproduced by a simple physically-based model proposed by Guarracino and Jougnot (2018), which consider a capillary as a first approximation for xylem vessel. It is seen that decreasing permeability leads to an increasing effective excess charge density, subsequently reducing electrical streaming current density (Eq. 3) and weakening the electrokinetic effect. Based on Eq. (12) and the $\hat{Q}_v$ estimates introduced in Section 4.3.2, we calculated the sapwood permeabilities for Aleppo pine (FBPh) as $4.49\times 10^{-13}$, $2.01\times 10^{-13}$, $1.06\times 10^{-13}$, and $5.38\times 10^{-13}$ m$^2$ in winter, spring, summer, and autumn, respectively. Detailed seasonal characteristics of two-week data for the Aleppo pine (FBPh) and Holm oak (FBQi) at the Font-Blanche site can be found in Table S2.

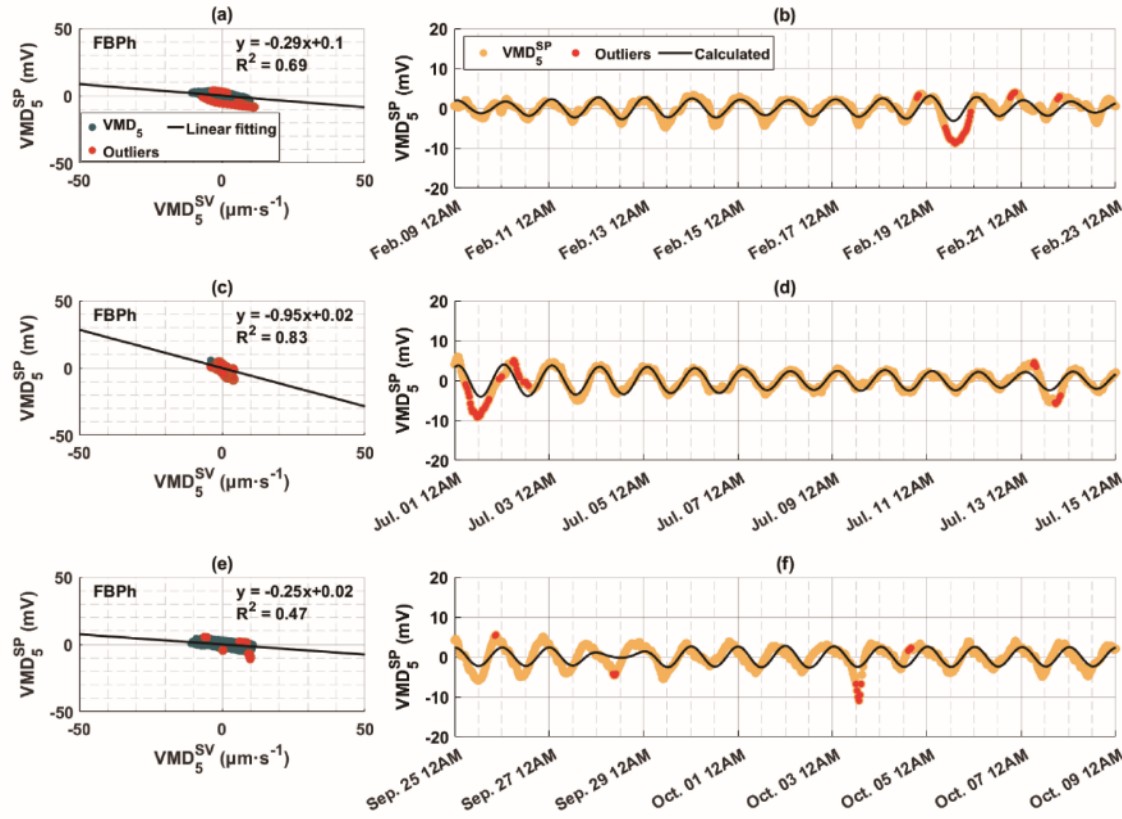

Figure 8: The relationship between the fifth decomposed modes of sap velocity ($VMD_5^{SV}$) and tree SP signals ($VMD_5^{SP}$) in (a–b) winter, (c–f) summer, and (e–f) autumn of 2023 for the Aleppo pine (FBPh) at the Font-Blanche. (a, c, e) Scatter plots of the decomposed modes of sap velocity and SP signals with black lines indicating linear regression results and red dots indicating points (outliers) outside the 95% confidence level of the regression. (b, d, f) Corresponding decompositions of SP signals (orange dots) and calculated SP (black lines) based on the linear relationship between $VMD_5^{SV}$ and $VMD_5^{SP}$, with red dots as outliers outside the 95% confidence level of the linear regression.

The ionic content and pH of xylem sap are reported to contribute to differences in hydraulic properties (López-Portillo et al., 2005; Losso et al., 2023). Referring to Eqs. (2) – (4) in Section 2.1 and assuming the electrical conductivity of xylem sap $\sigma_f$ is 270 µS·cm⁻¹ (Losso et al., 2023), and the dielectric permittivity of xylem sap $\epsilon_f$ is $7.10\times 10^{-10}$ F·m⁻¹ (equivalent to water at 20 °C), the corresponding Zeta potentials for FBPh are estimated to be –49.5 mV, –42.8 mV, –38.8 mV and –51.1 mV across the four seasonal periods, respectively. These Zeta potentials are of similar magnitude as typical values at the surface of silica grains in contact with an electrolyte, with concentration in the range of several millimoles per liter of NaCl (Revil et al., 1999). Although a lower sap rate is observed, a stronger electrokinetic coupling capacity in a tree still induces a larger amplitude SP. This suggests that the electrokinetic effects generated by sap flow might have similar characteristics to porous media. Thus, it has potential for short-term predictions using tree SP to evaluate sap velocity.

While the electrokinetic properties of plant tissues differ from those of mineral media due to their organic composition, similarities in surface charge behaviour and ion adsorption justify the preliminary use of porous media models to estimate SP and SV in plants. Plant tissues, such as xylem cell walls composed of cellulose, hemicellulose, lignin, and pectin, exhibit negatively charged surfaces of plasma membrane and carboxyl groups in pectin, allowing for cation binding. This behaviour is comparable to negatively charged mineral surfaces with electrical double layers. Studies have reported Zeta potentials in plant systems ranging from –48 mV to 23 mV, depending on pH and ionic conditions (Kinraide et al., 1998), which, while smaller in magnitude, remain comparable to those observed in silica grains. Although direct measurements of xylem Zeta potential are limited, research on lignocellulosic materials suggests they also exhibit negative Zeta potentials under near-neutral pH conditions (Hubbe, 2006). Despite these similarities, plant tissues fundamentally differ from mineral media, with their unique composition influencing water and ion transport. Experimental studies are essential to directly measure xylem Zeta potential and refine models for plant electrokinetic processes. This would improve the accuracy of SP and SV estimates and advance our understanding of electrokinetic phenomena in vascular plants.

As discussed in Section 2.1, several factors affect or alter tree SP. However, we contend that the gravitational forces driven by lunar-solar tides do not dominate tree SP generation as suggested in previous studies (Barlow 2012; Le Mouël et al., 2024). First, rainfall events do not change gravitational forces, yet tree SP was clearly affected by rainfall (e.g., Figs. 6f&h). Second, a strong link between diurnal tree SP and sap velocity is observed during the growing season, independent of rainfall (e.g., Figs. 4, 6a&b). While sap flow may not be the sole mechanism for tree SP generation, our results suggest a close relationship between tree SP and sap velocity. Further studies with monitoring of electrical resistivity, solute concentration, and water content in sapwood are needed to elucidate the mechanisms behind tree SP generation and variation.

**5.2 Electrochemical effects**

The results presented in Sections 4.3 indicate that electrokinetic mechanisms alone fail to fully explain the amplitudes of tree SP and the phase shifts between the diurnal variations of the measured SP and predicted SP from measured sap velocity data. Researchers propose that pH gradients contribute to SP differences (e.g., Gil and Vargas, 2023; Love et al., 2008). According to the Nernst potential:

$$\varphi_{\text{pH}}^{\text{lower}} - \varphi_{\text{pH}}^{\text{upper}} = -\frac{k_B T_K}{e_0 q_i}(\text{pH}^{\text{lower}} - \text{pH}^{\text{upper}}), \tag{13}$$

the electrochemical effect could dominate SP generation under significant pH contrasts between the measured electrodes; however, the electrokinetic effect associated with transpiration would still persist.

Schill et al. (1996) observed decreasing pH, ranging from 6.6 to 7.4, and an increase in osmotic potential with height in maple trees, with a pH gradient of approximately 0.8 m$^{-1}$ in the lower trunk. Applying Eq. (13) to a representative pH difference of $\Delta$pH= pH$^{\text{lower}}$ − pH$^{\text{upper}}$= 0.1 over 0.1 m yields a calculated potential difference of –2.53 mV at 20°C. Such pH-induced effects likely contribute to the underestimation of SP by sap velocity data (Figs. 6d, f, h). Furthermore, upward longitudinal concentration gradients of alkaline earth cations (K$^+$, Ca$^{2+}$, Mg$^{2+}$) in some tree stems (McDonald et al., 2002)

suggest complex sap chemistry, which may partly explain the non-linear relationship between SP and sap velocity observed in our study.

## 5.3 Experimental set-up concerning tree SP

In our measurements, stainless steel electrodes were employed to measure trunk SP. Compared to other materials such as copper or aluminum, stainless steel electrodes demonstrate relatively stable performance (Hao et al., 2015). However, it's essential to note that metal electrodes are sensitive to environmental changes and have a risk to be polarized. Although the distance between the tree electrodes is small, monitoring the temperature and adding two or three electrodes around each measured point can allow corrections for electrode drift induced by electrode-related effects, enabling a more accurate determination of the tree SP related to physiological changes.

As outlined earlier, all analyses conducted in Section 4 were performed without considering electrode effects. In future research, we suggest an improved experimental setup that involves distributing electrodes along the trunk with greater spacing (e.g., Fig. S11) and using duplicated measurements to ensure that the conclusions are robust. By enhancing the experimental setup, we will further strengthen our current understanding regarding the relationship between tree SP and the transpiration process.

Furthermore, the relationship between SP and upward sap flow does not systematically show a linear pattern. To explore the spatial characteristics associated with physiological processes, further observations and modelling are required. We recommend several directions for future research. First, expanding measurements to include multiple trees within the same site would improve the statistical robustness of findings and help clarify species-specific effects from general physiological principles. Second, utilizing ion-selective electrodes to assess the contributions of different ions to SP observations. Such measurements, combined with detailed analyses of xylem sap chemistry, would enable a more comprehensive understanding of the electrochemical effects driving SP signals. Third, we propose that future studies adopt a coupled experimental and modelling approach. Physiochemical models incorporating ionic transport, pH gradients, and electrokinetic effects could help quantify the relative contributions of these processes to observed SP patterns. Such models would benefit from controlled experiments under varying environmental conditions (e.g., soil moisture, temperature, and nutrient availability) to validate their predictive capabilities. With these advancements to refine SP monitoring potentially improve transpiration rate assessments, enable non-invasive studies of water transport dynamics, and contribute to our understanding of tree responses to environmental stressors.

## 6 Conclusions

This study presents a unique long-term dataset integrating SP and sap velocity measurements on four trees over a one-year period in a Mediterranean climate. Our findings reveal strong diurnal coherence between SP and sap velocity, which

weakens with higher water availability and is accompanied by increased phase shifts. Different trees, as well as the same tree across different seasons, exhibit distinct SP responses, with varying diurnal SP-to-sap velocity ratios.

During dry seasons, the strong agreement between observed SP and SP predicted from sap velocity data underscores the electrokinetic the dominance of electrokinetic effects. When electrokinetic mechanisms governed SP generation, we estimated the excess charge densities and Zeta potentials of Holm oak and Aleppo pine at the same site in different seasons using a linear regression model. The obtained values are comparable to the typical range for porous rock materials. Unlike point measurements of sap velocity, SP signals reflect an integrated bioelectrical process between the electrodes but are also influenced by electrode-related effects and environmental factors. These results highlight SP's potential as a tool for transpiration rate assessment in dry conditions. However, discrepancies during wet periods and non-linear relationships in some trees indicate additional physiological and electrochemical processes influencing SP signals.

While this study advances understanding of tree SP dynamics, certain limitations—such as short electrode separations and the use of single electrode pairs per tree—highlight key areas for future research. Enhancing electrode materials, optimizing experimental designs, and incorporating physiochemical modelling are important for further deciphering the complex bioelectrical processes governing SP signals. These advancements will improve the reliability of SP measurements and broaden their applications in ecohydrology and plant physiology.

**CRediT authorship contribution statement**

**Kaiyan Hu**: Conceptualization, Methodology, Fornal analysis, Data curation, Writing- Original Draft, Writing - Review & Editing, Visualization.

**Bertille Loiseau**: Investigation, Data curation, Writing - Review & Editing.

**Simon D. Carrière**: Investigation, Writing- Original Draft, Writing - Review & Editing.

**Nolwenn Lesparre**: Conceptualization, Writing - Review & Editing.

**Cédric Champollion:** Investigation, Writing - Review & Editing.

**Nicolas K. Martin-StPaul:** Supervision, Data curation.

**Niklas Linde**: Conceptualization, Methodology, Writing- Original Draft, Writing - Review & Editing, Supervision.

**Damien Jougnot**: Conceptualization, Methodology, Writing- Original Draft, Writing - Review & Editing, Project administration, Supervision.

**Declaration of competing interests**

The authors declare that they have no known competing financial interests or personal relationships that could have appeared to influence the work reported in this paper.

## Data availability

All data used in this study are freely accessible and available on Zenodo (https://doi.org/10.5281/zenodo.12662288) and will also be published on the H+ National Observatory Service with free access.

## Acknowledgements

The authors acknowledge the financial support of OZCAR Research Infrastructure, the H+ National Observatory Service and OSU OREME for funding part of this work. Nolwenn Lesparre, Damien Jougnot, and Simon Carrière also thank the MOOSE project, an EC2CO initiative. Kaiyan Hu thanks the financial support by the Postdoctoral Fellowship Program of China Postdoctoral Science Foundation (GZC20241598, 2024M753016) and the "CUG Scholar" Scientific Research Funds at China University of Geosciences (Wuhan) (Project No. 2023139).

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
