# Peer review of "Self-potential signals related to tree transpiration in a Mediterranean climate"

_Hydrology and Earth System Sciences, 2024_

## Author Comment (AC1)

Please find below our point-by-point response to the reviewer's comments. We quote all comments here in their entirety and all of our responses are in BLUE. All revisions are highlighted in blue color in the revised version of the manuscript.

My only few recommendations would be:

1. Remove reference to the excess charge densities of rocks when discussing this property interpreted for trees. Given the expertise of some authors this is not surprising, but I don't find this comparison relevant. It would be better to compare the data with similar.

   [**Response**] We will adjust the reference and its citation accordingly in the revised manuscript.

2. Add a brief description of the color scheme used in Figs. 4b-c. This could be done as a legend to the colored bar or a brief description in the caption.

   [**Response**] Thank the referee for this suggestion to enhance the figure's clarity. We have added a label indicating coherence to the color bar in Figs. 4b-c and provided a brief description in the caption to improve readability. Please refer to the updated figure below.

[Figure]

**Figure 4: Wavelet coherence analysis between sap velocity and tree SP data at the Font-Blanche site in 2023. (a) Water surplus calculated as the difference between precipitation and actual evapotranspiration; (b-c) Wavelet**

coherence maps for the Aleppo pine and Holm oak at the Font-Blanche site, respectively, with yellow highlighted boxes indicating periods of water surplus. Arrows denote the lag/lead phase between the two time series; White dashed lines (b-c) indicate the cone of influence where edge artifacts are negligible.

3. Sections 5.2 seems a bit too speculative, so I'd recommend to shorten the discussion and append it to Section 4. However, I will leave the decision on whether to do this or not to the authors.

    [**Response**] We appreciate the referee's suggestion. We will revise Section 5.2 to focus solely on describing the observed pattern of how pH differences influence SP amplitudes. The current content of Section 5.2 discusses the potential underestimation of predicted SP by sap velocity, as compared to the SP component derived from measured data. However, since we did not measure sap's pH values, this part is speculative. As the referee pointed out, Section 5.2 reads more like a hypothesis, so we will shorten this section rather than moving it to Section 4, while ensuring that the Results section remains focused on the solid findings derived from the actual data.

---

## Author Comment (AC2)

*Please find below our point-by-point response to the reviewer's comments. We quote all comments here in their entirety and all of our responses are in BLUE italics. All revisions will be highlighted in blue color in the revised version of the manuscript.*

**General comments**

This article presents very interesting data and data analysis of a 1 year monitoring of self potential (SP) and sap flow (SF) measurements on 3 different mediterranean forests. 1 or 2 trees are measured in each site, with single SP measurement on each tree. The paper is well written, well organized (except some parts, cf specific comments). It necessitates some preliminary skills in SP, but should be understandable for a broad audience. The strengths of the study are these long term measurements, the pertinent signal analysis used for understanding links (or no links) between SF and SP, which open way for understanding complicated signal from in situ experiments, and the quantifications with theoretical grounds for SP interpretation given by the authors. Exemplarily, the authors shows that it is not so much evident to relate SF and SP despite various studies started in the 60's and show the different processes that can affect the signal. If the strength of the study is in the diversity of site and trees, in my opinion there is a lack of discussion in the interpretation of the results in the discussion according to the different sites and tree species. From experimental point of view, only 1 tree is measured per site (2 at Font Blanche, but 1 tree of each species). If I understand that it could be a high workload, more replicated measured tree sampled would have been interesting, at least, to examine the range of variation for a same location. From another point of view, most of the interpreted results are based on 2 weeks in April for a 1 year study. May be looking at other contrasted periods or seasons would have been interesting for deciphering the different processes acting on SP signal (ie in winter when transpiration is slowed down or halted), in autumn when trees are recovering from possible drought, in summer during the drought and limitation of transpiration by water availability).

I add below some more specific comments, which can also call for some discussion on some points.

**[Response]** *We appreciate the reviewer's positive feedback regarding the data collection, analysis, and overall structure of the manuscript. We understand the concern about the limited number of measured trees per site. The selected trees in this work represent the dominant species at each location. Given the limited knowledge concerning tree SP, we agree with the reviewer that conducting repeated measurements on multiple trees at each site would enhance the understanding of SP characteristics.*

*However, we emphasize that this study presents a unique long-term dataset, integrating SP and sap velocity (SV) measurements across four trees. This dataset offers valuable insights into the relationship between SP and sap flow, as well as the influence of rainfall, over an extended period. By providing open access to this one-year dataset, we aim to encourage further analysis, measurements, and research into tree SP, fostering broader exploration in this relatively understudied area. Our findings highlight the electrokinetic effect as a primary driver of tree SP during dry seasons, demonstrating the method's potential for assessing transpiration rates. We believe this work may lay the foundation for future investigations into tree SP signals. In light of the reviewer's comments, we will expand the final paragraph of the Discussion section to recommend further SP measurements across multiple trees within the same site to enhance the* robustness of the

*findings.*

*We also appreciate the reviewer's suggestion to explore SP and SV dynamics during different seasons, particularly in winter, summer, and autumn when transpiration status are different. While our initial analysis focuses on a two-week period in April—representing active transpiration across different species (Figs. 3c&d)—we acknowledge the importance of seasonal variability in interpreting SP signals. To address this, we have extended our analysis to include data from winter, spring, summer, and autumn, selecting periods without rainfall to avoid confounding effects (Figs. R1-R2).*

*The expanded analysis indicates that SP signals remain responsive throughout the year, albeit with variations in intensity. Increased phase shifts between SV and SP are observed during cooler months (Fig. R1b&h, R2b&h), yet the predicted SP derived from SV data aligns well with decomposed SP observations. Notably, the linear relationship between SV and SP is more pronounced for Aleppo pine compared to Holm oak (Table R1). For both species, SP correlates strongly with sap flow during spring and summer. The slope of the linear model between decomposed SV and SP data peaks in summer (Fig. R1e, R2e), reflecting heightened SP amplitudes despite indications of drought-induced limitations in SV (Figs. R1e&f, R2e&f). Additionally, the raw SV data exhibits an evident decline in July (Fig. 3c-d).*

*We recognize that this dataset has limitations, as measurements were conducted on individual trees. A larger sample size would provide more robust insights into inter-tree variability and seasonal patterns. Furthermore, while this study focuses primarily on the electrokinetic effects related to transpiration, we acknowledge the potential influence of other physiological processes. Investigating SP generated by alternative mechanisms, such as ion movement and concentration changes, could yield valuable insights if additional data representing these processes were collected. However, the current dataset primarily includes SV and SP measurements (Fig. 2). As such, we lack direct evidence linking SP to processes beyond transpiration.*

*In response to the reviewer's comment, we will revise the Discussion section to discuss the seasonal relationships between SP and SV and acknowledge the need for future studies exploring other physiological drivers of SP.*

[Figure]

*Figure R1: The relationship between the fifth decomposed modes of sap velocity ($VMD_5^{SV}$) and tree SP ($VMD_5^{SP}$) signals in (a-b) winter, (c-d) spring, (e-f) summer, and (g-h) autumn of 2023 for the Aleppo pine (FBPh) at the Font-Blanche site. (a, c, e, f) Comparison scatter plots of the decomposed modes of sap velocity and SP signals with black lines indicating linear regression results and red dots indicating points (outliers) outside the 95% confidence level of the regression. (b, d, f, h) Corresponding decompositions of SP signals (orange dots) and calculated SP (black lines) based on the linear relationship between $VMD_5^{SV}$ and $VMD_5^{SP}$, with red dots as outliers outside the 95% confidence level of the linear regression.*

[Figure]

*Figure R2: The relationship between the fifth decomposed modes of sap velocity ($VMD_5^{SV}$) and tree SP ($VMD_5^{SP}$) signals in (a-b) winter, (c-d) spring, (e-f) summer, and (g-h) autumn of 2023 for the Holm oak (FBQi) at the Font-Blanche site. (a, c, e, f) Comparison scatter plots of the decomposed modes of sap velocity and SP signals with black lines indicating linear regression results and red dots indicating points (outliers) outside the 95% confidence level of the regression. (b, d, f, h) Corresponding decompositions of SP signals (orange dots) and calculated SP (black lines) based on the linear relationship between $VMD_5^{SV}$ and $VMD_5^{SP}$, with red dots as outliers outside the 95% confidence level of the linear regression.*

*Table R1 Seasonal characteristics of two-week data for the Aleppo pine (FBPh) and Holm oak (FBQi) at the Font-Blanche site.*

| Season | Property | | FBPh | FBQi |
|--------|----------|--|------|------|
| **Winter** | Range | $VMD_5^{SP}$ (mV) | [−4.78, +3.90] | [−21.81, +17.83] |
| | | $VMD_5^{SV}$ (μm·s⁻¹) | [−10.56, +9.86] | [−5.61, +6.49] |
| | Standard Deviation | $VMD_5^{SP}$ (mV) | 1.89 | 7.13 |
| | | $VMD_5^{SV}$ (μm·s⁻¹) | 5.49 | 3.00 |

| | | | | |
|---|---|---|---|---|
| | $\hat{Q}_v$ (C·m$^{-3}$) | | 7.9 | 43.2 |
| | Correlation Coefficient | | 0.83 | 0.45 |
| **Spring** | Range | VMD$_5^{SP}$ (mV) | [−5.57, +4.69] | [−19.52, +16.04] |
| | | VMD$_5^{SV}$ (μm·s$^{-1}$) | [−6.29, +6.36] | [−10.60, +10.61] |
| | Standard Deviation | VMD$_5^{SP}$ (mV) | 2.28 | 7.51 |
| | | VMD$_5^{SV}$ (μm·s$^{-1}$) | 3.61 | 6.33 |
| | $\hat{Q}_v$ (C·m$^{-3}$) | | 15.2 | 31.2 |
| | Correlation Coefficient | | 0.88 | 0.66 |
| **Summer** | Range | VMD$_5^{SP}$ (mV) | [−5.58, +5.93] | [−26.82, +22.35] |
| | | VMD$_5^{SV}$ (μm·s$^{-1}$) | [−4.22, +4.09] | [−9.36, +9.32] |
| | Standard Deviation | VMD$_5^{SP}$ (mV) | 2.24 | 11.94 |
| | | VMD$_5^{SV}$ (μm·s$^{-1}$) | 2.15 | 5.43 |
| | $\hat{Q}_v$ (C·m$^{-3}$) | | 25.7 | 68.0 |
| | Correlation Coefficient | | 0.91 | 0.77 |
| **Autumn** | Range | VMD$_5^{SP}$ (mV) | [−5.94, +5.06] | [−14.73, +11.81] |
| | | VMD$_5^{SV}$ (μm·s$^{-1}$) | [−10.90, +10.89] | [−4.48, +4.68] |
| | Standard Deviation | VMD$_5^{SP}$ (mV) | 2.44 | 6.68 |
| | | VMD$_5^{SV}$ (μm·s$^{-1}$) | 6.68 | 2.82 |
| | $\hat{Q}_v$ (C·m$^{-3}$) | | 6.8 | 51.6 |
| | Correlation Coefficient | | 0.69 | 0.54 |

*Below is our detailed point-by-point response to the reviewer's comments.*

**Specific comments**

13 Transpiration is => Plant transpiration is

**[Response]** *We thank the reviewers for this addition. We will revise the text to specify "Plant transpiration".*

14 Solely relying on sap flow measurements => Not really as at the stand scale (evapo) transpiration

can also be assessed by flux towers' measurements.

**[Response]** *We generally agree with the reviewer, but our focus here is on techniques for measuring transpiration at the individual tree level.*

*Flux towers, while effective for estimating ecosystem-scale evapotranspiration, do not directly measure transpiration from individual trees. Instead, flux towers capture the combined water loss from soil evaporation and plant transpiration, typically requiring additional modeling or partitioning techniques to isolate transpiration (e.g., Nelson et al., 2020). In contrast, sap flow measurements provide a direct method to quantify transpiration at the individual tree level. To clarify this point, we will revise this sentence with "While sap flow measurements offer a direct method for estimating individual tree transpiration, their effectiveness may be limited by the use of point sensors, which may not fully capture whole-tree dynamics".*

*Reference:*

*Nelson, J. A., Pérez-Priego, O., Zhou, S., Poyatos, R., Zhang, Y., Blanken, P. D., ... & Jung, M. (2020). Ecosystem transpiration and evaporation: Insights from three water flux partitioning methods across FLUXNET sites. Global Change Biology, 26(12), 6916-6930.*

32-37 Again cite different methods for (evapo)transpiration measurement: flux towers, soil water balance. With the specifity of sap flow measurement being the direct method for trees for obtaining transpiration.

**[Response]** *We thank the reviewer for this suggestion. In this section, we focus on transpiration measurements at the individual tree level rather than stand-scale evapotranspiration. We will clarify that there are three primary methods for measuring transpiration. We will revise the text to clarify it with "Plant transpiration can be measured using various techniques, including sap flow methods that quantify water movement through the xylem (e.g., Goulden et al., 1994; Granier et al., 1996; Kume et al., 2010), porometry to assess leaf-level transpiration and stomatal conductance (e.g., Zhang et al., 1997; Damour et al., 2010), and flux towers (eddy covariance) that estimate stand-scale evapotranspiration, with tree transpiration inferred by partitioning soil evaporation (e.g., Kurpius et al., 2003; Scanlon and Kustas, 2012)".*

*Goulden, M. L., & Field, C. B. (1994). Three methods for monitoring the gas exchange of individual tree canopies: ventilated-chamber, sap-flow and Penman-Monteith measurements on evergreen oaks. Functional Ecology, 125-135.*

*Granier, A., Biron, P., Bréda, N., Pontailler, J. Y., & Saugier, B. (1996). Transpiration of trees and forest stands: Short and long-term monitoring using sapflow methods. Global Change Biology, 2(3), 265-274.*

*Zhang, H., Simmonds, L. P., Morison, J. I., & Payne, D. (1997). Estimation of transpiration by single trees: comparison of sap flow measurements with a combination equation. Agricultural and Forest Meteorology, 87(2-3), 155-169.*

*Kurpius, M. R., Panek, J. A., Nikolov, N. T., McKay, M., & Goldstein, A. H. (2003). Partitioning of water flux in a Sierra Nevada ponderosa pine plantation. Agricultural and Forest Meteorology, 117(3-4), 173-192. https://doi.org/10.1016/S0168-1923(03)00062-5*

*Damour, G., Simonneau, T., Cochard, H., & Urban, L. (2010). An overview of models of stomatal*

*conductance at the leaf level. Plant, Cell & Environment, 33(9), 1419-1438. https://doi.org/10.1111/j.1365-3040.2010.02181.x*

*Kume, T., Onozawa, Y., Komatsu, H., Tsuruta, K., Shinohara, Y., Umebayashi, T., & Otsuki, K. (2010). Stand-scale transpiration estimates in a Moso bamboo forest:(I) Applicability of sap flux measurements. Forest Ecology and Management, 260(8), 1287-1294.*

*Scanlon, T. M., & Kustas, W. P. (2012). Partitioning evapotranspiration using an eddy covariance-based technique: Improved assessment of soil moisture and land–atmosphere exchange dynamics. Vadose Zone Journal, 11(3). https://doi.org/10.2136/vzj2012.0025*

42 Gindl et al., 1999;) => suppress last;

**[Response]** *We thank the reviewer for pointing out this typo. The semicolon will be removed in the revised version.*

46 transpiration processes, which facilitate water and solute transport within the xylem and phloem of trees => Rather : transpiration which relies on the transport of water and solutes in xylem…. Phloem transport is not generated by transpiration but by gradients of concentration of sugars which generate turgor pressure gradients and flow, and can occur without transpiration. So add another sentence, for phloem, decoupled from transpiration

46 , trigger electrokinetic and electro-diffusive effects, => a few words to explain the origin of effects ?

**[Response]** *We thank the reviewer for these two comments. We will reword this part with "Tree transpiration processes facilitate the transport of water and solutes within the xylem (e.g., Kim et al., 2014). Additionally, sugar concentration gradients can generate turgor pressure differences, driving flow within the phloem (e.g., van Bel, 2003). These natural processes can trigger electrokinetic effects through the advection of net electrical charges and electro-diffusive effects driven by electrochemical potential gradients, leading to the generation of biopotentials and measurable SP signals".*

*Reference:*

*van Bel, A. J. (2003). Transport phloem: low profile, high impact. Plant Physiology, 131(4), 1509-1510. https://doi.org/10.1104/pp.131.4.1509*

*Kim, H. K., Park, J., & Hwang, I. (2014). Investigating water transport through the xylem network in vascular plants. Journal of Experimental Botany, 65(7), 1895-1904. https://doi.org/10.1093/jxb/eru075*

132 low of sap induces a natural electric field in the opposite direction, as depicted in Fig. 1e => no clear indication of opposite electric field in 1e ? Only in the transverse direction of the vessel with the subplot. Could you make the graph clearer for this electric potential difference generation.

**[Response]** *In Fig. 1e, the direction of sap flow is indicated by the upward arrow on the right side, while the downward arrow on the left side represents the direction of the electric field generated by the streaming current. This visualization aligns with the electrical double layer model, where an excess of electric charges exists within the Gouy-Chapman diffuse layer. As sap flow drags these excess positive charges upward, it induces an upward streaming current ($J_{ek}$), resulting in the generation of a downward electric field (Eq. 1). We will add a sentence to clarify the electrokinetic*

*mechanism with "For negatively charged cell walls, the streaming current, generated by the upward movement of excess positive charge within the Gouy-Chapman diffuse layer as sap flows, represents the advective transport of electrical charges, resulting in a net source of current density $J_{ek}$".*

134 no external currents (i.e., electrical flux equals 0). => what means external current here? If there if a flow of ion charges what is the return current so that net current J=0 ?

**[Response]** *In this context, "external currents" refer to electrical currents generated by sources other than the electrokinetic effect, such as externally injected currents. The phrase "electrical flux equals 0" indicates that the current density produced by the electrokinetic effect is counterbalanced by a conduction current density, maintaining charge conservation and resulting in no net external current.*

*To enhance clarity, we will revise the sentence to:*
*"This describes a condition where only electrokinetic effects are present, with no contribution from diffusive currents in Eq. 1 or any external current sources, ensuring that the net current remains zero."*

137 of the capillary fluid => flowing fluid ?

**[Response]** *Yes, it will be revised to "flowing fluid" here.*

140  an empirical relationship => for which media ?

**[Response]** *We will complete this sentence to:"...Linde et al., (2007) proposed an empirical relationship between the logarithm of $C_{ek}$ and $\sigma_f$ for porous media".*

147 where k (m2) denotes permeability => for a porous medium?

**[Response]** *We will revise this sentence to "...k ($m^2$) denotes permeability for a porous medium".*

148 under the assumption of 1-D flow,  => and neglecting Jdiff  in (1)

**[Response]** *As suggested by the reviewer, we will revise this sentence to "...under the assumption of 1-D flow and neglecting $J_{diff}$ in Eq. 1, the current density is solely governed by the electrokinetic effect".*

149- 155 lead to an increase in solute concentration towards the crown => If available, give range of variation in concentration, that should be small as ions are absorbed by cells too..

**[Response]** *The xylem sap typically contains approximately 10 mM of inorganic nutrients, along with organic nitrogen compounds metabolically synthesized in the root (Nobel, 2009). According to McDonald et al. (2002), for a Norway spruce tree in late autumn, the total concentration of amino acids and the minerals magnesium, calcium, and potassium in the xylem sap ranges from 4 mM at the ground level to 6 mM near the crown base. This concentration range aligns with typical salinity levels in pore water (Linde et al., 2007). We will add the range of variation in sap concentration to the corresponding text.*

Table 1 Soil depth is indicated in m (range 0-80 m !), it is rather cm I think  …   Evapotranspiration seems low, if potential. Is it actual or potential evapotranspiration ?

**[Response]** *We thank the reviewer for pointing out this typo. The unit of soil depth is centimeter indeed. The evapotranspiration is "actual evapotranspiration", which will be clarified in the revised version.*

L240 the sap velocity in μm s => the sap velocity is obtained in μm s-1

[Response] *This sentence will be corrected in the revised manuscript.*

L243 high-impedance multimeter controlled by a digital data logger => give the reference of manufacturer and model. What is the sampling frequency of recorded voltages?

[Response] *The sampling intervals for tree SP measurements are 1 minute at the Larzac and LSBB sites, and 10 minutes at the Font-Blanche site. Detailed measurement parameters, including sampling intervals and units, are provided in Table S1.*

*We will also include additional information about the equipment used for the measurements. The high-impedance multimeter, with an input impedance of 20 GΩ, was controlled by a Campbell Scientific CR1000X datalogger.*

L244 which length and diameter for screws? Fig 2 it seems it is not screws for wood works. True? Does screws penetrate sapwood or are just in contact with first outside layer of vessels?

[Response] *The size of the trunk electrode is shown in Fig. 2f, and the electrodes used are 3 mm diameter screws made of stainless steel. As the bark was peeled from the trunk prior to electrode installation (similarly to the installation of the rod of sapflow meters), the screws penetrate the sapwood to a depth of 2 cm beneath the cork cambium (Line 245).*

L272 In addition to variational mode decomposition, would Fourier spectrum analysis give also similar or complementary results?

[Response] *As illustrated in Fig. R3, Fourier spectrum analysis of sap velocity (SV) data reveals similar dominant frequencies to those identified through variational mode decomposition (VMD) (Fig. 5). Both methods indicate the presence of 1-, 2-, 3-, and 4-day$^{-1}$ frequency signals in the SV data.*

*However, as noted in Lines 273-274, SP measurements are more complex. They are susceptible to high-frequency electromagnetic noise, temperature fluctuations, trunk wounds, and electrode polarization, which can introduce long-period drifts and complicate the signal composition. Consequently, the SP spectrum (Figs. R3c-d) does not exhibit clear dominant frequencies that can be directly characterized through Fourier analysis.*

*This is the primary reason we employed wavelet transformation and VMD to analyze the data, as these methods are more effective in isolating multiple signals from the SP data. To clarify this point, we will revise the text accordingly and include Fig. R3 in the Supplementary Material to explain why VMD was chosen for this study.*

[Figure]

*Figure R3: Magnitude of the sap velocity (SV) and self-potential (SP) spectra using a fast Fourier transform algorithm. (a, c) SV and SP spectra for Aleppo pine (FBPh). (b, d) SV and SP spectra for Holm oak (FBQi) at the Font-Blanche site.*

Fig 3 The time axis is labelled as Jan.01, Feb.01.. May be Jan 23, Feb 23… Jan 24 would be better.   30minutes data are shown, daily values would be interting to show (cumulative as proxy of daily flow and average for SP probably)

**[Response]** *We appreciate the recommendation regarding the time axis labeling. However, we believe that displaying the first day of each month (e.g., Jan. 01, Feb. 01) provides a clear and consistent representation of the time series. Therefore, we prefer to retain the current time axis labeling.*

*In line with your suggestion, we have added the daily average of SP data and the daily cumulative sap flow to Figs. 3c-f to provide a clearer visualization of trends and variations over time.*

[Figure]

*Figure R4 (Figure 3 in the manuscript): One-year data collected at the Font-Blanche site at half-hourly intervals from January 1, 2023, to January 1, 2024. (a) Precipitation and air temperature data; (b) Actual evapotranspiration (Actual ET) and vapor pressure deficit (VPD) data; (c-d) Measured sap velocity in blue lines and daily cumulative sap flow in pink lines for the Aleppo pine (FBPh) and the Holm oak (FBQi), respectively; (e-f) Measured SP data in blue lines and daily average of SP data in pink lines for the Aleppo pine (FBPh) and the Holm oak (FBQi), respectively.*

L306 coherence anomalies => Why anomalies ? not just the coherence, the highest at 1 day, and meaning, I think, that the SP and SV are more linked in terms of variations to each at a 1 day period than other periods.

**[Response]** *We acknowledge that the term "anomalies" may lead to confusion. To improve clarity, we will revise the sentence as follows:*

*"In particular, the one-day period shows high coherence, with arrows pointing left, indicating that SP is negatively correlated with SV at this timescale. This indicates that SP and SV exhibit stronger correlation at a daily timescale compared to other periods."*

L307 negative correlation => ?, figure 4 is not correlation. If you refer to linear correlation in S1, put this sentence before to show the difference in data interpretation between wavelet and correlation.

**[Response]** *Wavelet coherence is indeed a localized correlation in time-frequency space, as described for instance by Grinsted et al. (2004). The coherence maps in Figs. 4b-c illustrate the temporal correlation between SV and SP across different time periods. The arrows in Figs. 4b-c represent the phase relationship between SV and SP. Particularly, the rightward arrows (0° phase) indicate in-phase behavior, where SV and SP vary synchronously. Leftward arrows (180° phase)*

*indicate anti-phase behavior, signifying negative correlation or out-of-phase variations between SV and SP. Conversely, upward or downward arrows (±90° phase) reflect phase shifts, suggesting that one signal leads or lags the other by a quarter cycle. In particular, the one-day period shows high coherence, with arrows pointing left, indicating that SP is negatively correlated with SV at this timescale. To improve clarity, we will add a few sentences to Section 4.1.2 explaining the phase relationships and the significance of the arrows in Fig. 4.*

*Reference:*

*Grinsted, A., Moore, J. C., & Jevrejeva, S. (2004). Application of the cross wavelet transform and wavelet coherence to geophysical time series. Nonlinear Processes in Geophysics, 11(5/6), 561-566. 1607-7946/npg/2004-11-561*

Fig4 the meaning for arrows as lag phase is not very clear to me. It is not explained and used in the text in 4.1.2. If this data it is needed it should be explained and used.

**[Response]** *As our response to the above comment, we will add a few sentences as follows:*

*"The arrows in Figs. 4b-c represent the phase relationship between SV and SP. Particularly, the rightward arrows (0° phase) indicate in-phase behaviour, where SV and SP vary synchronously. Leftward arrows (180° phase) indicate anti-phase behaviour, signifying negative correlation or out-of-phase variations between SV and SP. Conversely, upward or downward arrows (±90° phase) reflect phase shifts, suggesting that one signal leads or lags the other by a quarter cycle."*

L324 330 Can you comment not only on frequency differences but also on the "power" of signal associated to those frequencies?

**[Response]** *To address the reviewer's comment, we will expand the description to discuss not only the frequency differences but also the spectral amplitudes associated with those frequencies.*

*We will revise the section as follows:*

*"… noise present in the tree SP data tends to manifest at higher frequencies, as indicated by the prominent spectral amplitudes in the higher-frequency range (Figs. 5a, c, e, g). The low-frequency components, which reflect baseline trends, dominate the last intrinsic modes and occupy the largest proportion of the signals, as shown by the highest amplitudes in the SP and SV spectra (Figs. 5k-l). Apart from the fifth intrinsic mode, the decompositions of SP and SV exhibit distinct frequency characteristics, highlighting the complexity of SP signals. Notably, the fifth intrinsic modes for both datasets reveal a strong diurnal rhythm, with a dominant frequency of $1\ d^{-1}$ (Figs. 5i-j). The spectra of the fifth decomposition also display the second-highest amplitudes, indicating that these diurnal components represent a significant portion of the overall SP and SV signals"*

Fig 5: x axis could be labelled Frequency (d-1), for better understanding. Indicate what means |u1(f)| … in the legend.

**[Response]** *Done.*

[Figure]

*Figure R5 (Figure 5): Frequency spectra of six decomposed modes of tree SP (left column: a, c, e, g, i, k) and sap velocity (left column: b, d, f, h, j, l) data obtained on the Holm oak (FBQi) at the Font Blanche site within 2023 using the variational method (VMD); Different rows correspond to different modes, where "CF" and "DF" indicate the center frequency and dominant frequency of the corresponding mode, respectively.*

349 during a two-week period in the growing season => Which 2 weeks and why those 2 weeks?

**[Response]** *The two-week period from April 16 to April 30 was selected to capture a phase of active transpiration across different species, during which no precipitation was recorded (as shown in Fig. 3). We will add a sentence to clearly state the rationale for selecting this timeframe.*

L352 – 354 Negative correlation, and flow direction. If OK for the 5th mode of SV, looking at raw data on fig 3, SP of pine is negative, while for oak is positive … (and would lead to a positive and negative correlation), how to reconciliate with your statement?

**[Response]** *The raw SP data for both Aleppo pine and Holm oak indeed show fluctuating trends that span both negative and positive ranges (Figs. 3e-f). This variability arises primarily from long-term electrode drifts and low-frequency influences unrelated to sap flow, as mentioned in Lines 273-274. These drifts can obscure the intrinsic relationship between SV and SP. To address this, we applied wavelet analysis and VMD to isolate SP signals corresponding to the dominant sap flow frequencies. By focusing on the fifth decomposed mode, which shares the same diurnal frequency as sap flow, we effectively removed low-frequency drift and high-frequency noise, allowing us to focus on the intrinsic relationship between SP and SV.*

*The coherence maps in Fig. 4, where most phase arrows point left, indicate negative correlations between SP and SV in the one-day period. Additionally, the negative ratios of the fifth decomposed*

*modes of SP to SV for both Aleppo pine and Holm oak (Fig. 6a&c) further confirm this negative relationship.*

*We will revise the manuscript to include the following clarification:*
*"The large long-term fluctuations in SP data across positive and negative ranges, as well as high-frequency interferences, are excluded by extracting the fifth decomposed mode. This allows us to focus on relative variations aligned with the dominant sap flow frequency, providing clearer insight into the relationship between SP and SV."*

L359 and others occurrence in the text : The fifth decomposed signals => rather the fifth decomposed mode of signals. In the text, and fig legend (eg in fig 6) it could be clearer if you use "the fifth decomposed mode of" …

**[Response]** *We are thankful for the reviewer's suggestion. The "the fifth decomposed signals" will be replaced with "the fifth decomposed mode of". Correspondingly, the caption of Fig. 6 is revised as follow:*

*"Figure 6: The relationship between the fifth decomposed modes of sap velocity ($VMD_5^{SV}$) and tree SP ($VMD_5^{SP}$) signals in April 16-30, 2023, at the Font-Blanche, LSBB, and Larzac site. (a, c, e, f) Comparison scatter plots of the fifth decomposed modes of sap velocity and SP signals for (a) the Aleppo pine (FBPh), (c) the Holm oak (FBQi) at the Font-Blanche site, (e) the Holm oak at the LSBB (LSQi), and (g) the Pubescent oak at the Larzac (LaOp), with black lines indicating linear regression results and red dots indicating points (outliers) outside the 95% confidence level of the regression. (b, d, f, h) Corresponding decompositions of SP signals (orange dots) and calculated SP (black lines) based on the linear relationship between $VMD_5^{SV}$ and $VMD_5^{SP}$, with red dots as outliers outside the 95% confidence level of the linear regression."*

L361 applying linear regression => regression of SP vs SV

**[Response]** *This sentence will be completed in the revised version.*

L360- 364 In Fig 6, the 5th mode of SV shows negative values (almost half of data?), ie negative sap flow. What does that mean from a plant point of view?

**[Response]** *The negative values observed in the fifth decomposed mode of SV (Fig. 6) primarily reflect relative variations around the baseline of signals, rather than indicating actual reverse or negative sap flow. This mode captures diurnal fluctuations and oscillations but does not retain the baseline component, which is represented by the sixth decomposed mode. As a result, negative values in the fifth decomposed mode signify periods of lower sap flow relative to the baseline and not negative flow in a physiological sense.*

*To clarify this point, we will add the following sentence to the text:*
*"Please note that the negative values in the fifth decomposed mode of SV signals represent deviations from the baseline, reflecting diurnal fluctuations rather than indicating actual reverse sap flow. The baseline component is captured in the sixth decomposed mode, isolating physical sap flow from the oscillatory variations seen in the fifth decomposed mode."*

Fig 6 in legend: red dots indicating outliers => red dots indicating points (outliers) outside the 95% confidence level of the regression. In the graph => In blue Prec. could b replaced by rain. There is no rains at Font Blanche? Why there is blue dots on graphs (b,d,f…). Indicate r² of correlations (a, c,…)

**[Response]** *To address the comments:*

1. *Rainfall at Font-Blanche:*
   *We confirm that no precipitation occurred at the Font-Blanche site during the analyzed period. To clarify this, we have added the following sentence to Section 4.3.1:*
   *"The two-week period from April 16 to April 30 was selected to capture a phase of active transpiration across different species, during which no precipitation was recorded at the Font-Blanche site (as shown in Fig. 3)."*

2. *Labeling of Precipitation (Prec.):*
   *Since precipitation data is presented on the right y-axis in Figs. 6b, d, f, h, we prefer to retain the "Prec." label to clearly denote precipitation values.*

3. *Blue Dots in Graphs (b, d, f, h):*
   *The blue dots represent the edge color of the fifth decomposed mode of SP signals (orange dots). To avoid confusion, we have removed the edge color from the dots, ensuring the focus remains on the primary data points.*

4. *Outliers and Regression:*
   *We revised the caption to:*
   *"Red dots indicate points (outliers) outside the 95% confidence level of the regression."*

5. *$R^2$:*
   *In line with the reviewer's suggestion, we have added $R^2$ values to Figs. 6a, c, e, g to provide clearer information about the strength of the correlations.*

*Please see the revised Fig.6 as follow:*

[Figure]

*Figure R6 (Figure 6): The relationship between the fifth decomposed modes of sap velocity ($VMD_5^{SV}$) and tree SP ($VMD_5^{SP}$) signals in April 16-30, 2023, at the Font-Blanche, LSBB, and Larzac site. (a, c, e, f) Comparison scatter plots of the fifth decomposed modes of sap velocity and SP signals for (a) the Aleppo pine (FBPh), (c) the Holm oak (FBQi) at the Font-Blanche site, (e) the Holm oak at the LSBB (LSQi), and (g) the Pubescent oak at the Larzac (LaQp), with black lines indicating linear regression results and red dots indicating points (outliers) outside the 95% confidence level of the regression. (b, d, f, h) Corresponding decompositions of SP signals (orange dots) and calculated SP (black lines) based on the linear relationship between $VMD_5^{SV}$ and $VMD_5^{SP}$, with red dots as outliers outside the 95% confidence level of the linear regression.*

L364 383: All of this text should be put either at the end of the section, or a new result section created such as "excess charge density estimation". Which u is used here? from sap flow or 5th mode of sap flow? Why no calculation for other sites?

**[Response]** *As recommended, we will relocate this section to the end of Section 4.3.2 and entitle Section 4.3.2 as "Relationship between tree SP and sap velocity across four trees".*

*To clarify the notation, the original **u** refers to the fifth decomposed model of sap velocity signals. To avoid confusion with sap flux or the symbols used in Fig. 5, we will replace **u** with **v**. Here, $v \approx VMD_5^{SV}$ represents localized (relative) sap velocity derived from the fifth decomposed mode of the SV signals. We reword this sentence for clarity as follows: "By converting the localized $\nabla V \approx -\frac{VMD_5^{SP}}{10\ cm}$ and $v \approx VMD_5^{SV}$ to the International System of Units, we estimated the effective excess charge density ($\hat{Q}_v = \frac{\sigma \nabla V}{v}\Big|_{J=0}$) using linear fitting slope coefficients $\left(\frac{\nabla V}{v}\right)$".*

*Regarding the absence of calculations for other sites, the following factors influenced this decision: (1) No resistivity measurements were conducted for Holm oak (LSQi) at LSBB (as noted in Table 1); (2) Both the Larzac and LSBB sites experienced rainfall during the selected periods, resulting in poor correlations between SV and SP (Figs. 6e-h). This indicates that sap flow did not primarily drive SP generation at these locations, limiting the applicability of excess charge density estimation.*

*To clarify this in the manuscript, we will add the following sentence:*
*"Due to the lack of resistivity measurements at LSBB (LSQi) and the presence of rainfall at the Larzac and LSBB sites during the selected periods, excess charge density calculations were limited to the two trees at the Font-Blanche site, where sap flow consistently dominated SP generation."*

L391 – 398 and fig 7 : All of this text is difficult to understand. What is actually done is a difficult to follow, and figure 7 is not well explained…. Could it be more clearly rewritten?

**[Response]** *We will revise the text to improve readability and provide a clearer explanation of the analysis performed as*
*"Since the amplitude of the calculated SP (derived from the fifth decomposed mode of SV signals, $VMD_5^{SV}$) is lower than that of the fifth decomposed mode of SP signals ($VMD_5^{SP}$) for Holm oak at the Font-Blanche site on April 21–22, and 24 (Fig. 6d), we conducted a test to investigate whether the VMD method reduces the relative amplitude of SV signals during decomposition. To achieve this, we compared the raw SP and SV data to the sum of the diurnal features ($VMD_5^{SP}, VMD_5^{SV}$) and the baseline component ($VMD_6^{SP}, VMD_6^{SV}$) obtained from the VMD-processed data (Figs. 7a-b). The combination of the fifth and sixth decomposed modes is analogous to low-pass filtering, retaining lower-frequency signals. The highest diurnal amplitudes of the raw and processed SV data remain relatively stable, whereas SP data shows notable variations, particularly on April 21–22 and 24. To further investigate whether the underestimation of SP using $VMD_5^{SV}$ resulted from the exclusion of high-frequency components, we examined the sum of the first to fifth decomposed modes of SV signals, representing a detrending process. In Fig. 7c, the fifth decomposed mode of SP ($VMD_5^{SP}$) was compared with the SP calculated from $\sum_{i=1}^{5} VMD_i^{SV}$. This comparison demonstrated that even when all decomposed modes except the baseline were used to reconstruct diurnal SP from SV data, the higher SP amplitudes observed on April 21–22 and 24 could not be reproduced (Fig. 7c). To corroborate this observation, we repeated the analysis using a finite impulse response (FIR) filter, which yielded similar results (Fig. 7d). This suggests that the elevated SP magnitudes observed on these dates may be influenced by factors other than sap flow."*
*The caption will be modified to enhance the clarity of this test:*
*Figure 7: Two-week SP data during the 2023 growing season for Holm oak (FBQi) at the Font-Blanche site processed using different approaches. (a) Raw SP data compared with the sum of the last two decomposed models of SP signals; (b) Raw sap velocity data compared with the sum of the last two decomposed models of SV signals; (c) Fifth decomposed mode of SP signals, alongside SP calculated from the sum of the first to fifth decomposed modes of SV signals ($\sum_{i=1}^{5} VMD_i^{SV}$); (d) SP data filtered using a finite impulse response (FIR) filter (bandpass: 1/48 h⁻¹ to 1/1.5 h⁻¹; filter order: 100) compared with SP calculated from FIR-filtered SV data using the same filter parameters.*

L413-415 analogy between unsaturated soil and plant hydraulic conductivity seems doubtful to me. In soil, variation in water content is linked to filling/emptying of pores? Water content in plants is expressed generally on a (fresh) weight basis, not volumetric. When plant desiccates its volume change but not necessarily saturation. Indeed the plant variation in water content is rather loss of

water from cells, with a loss of turgor, and not from xylem which remains full of water except in very dry situations when embolism happens. If it could be the case in dry summer, it might not be that for spring, autumn? At least that should be discussed in such analogy.

**[Response]** *We acknowledge that the analogy between unsaturated soil and plant hydraulic conductivity requires clarification, as the mechanisms governing water movement in xylem differ from those in soils.*

*In soils, changes in water content are primarily driven by the filling or draining of pores, leading to variations in saturation. In contrast, the upward water movement in plants is largely confined to the xylem, which typically remains water-filled under normal conditions. Water loss in plants generally occurs at the cellular level, resulting in reduced turgor pressure rather than desaturation of the xylem. Xylem desaturation and embolism occur primarily during extreme drought, which is more common in dry summer conditions. While the dynamics of water movement in sapwood differ from those in the vadose zone, certain functional similarities exist in the flow in xylem and soil could be described by Poiseuille's Law (e.g., Nobel, 2009; Jougnot et al., 2019), and the functional response of hydraulic conductivity to water potential gradients (Cai et al., 2022).*

*To address this, we will revise the manuscript to clarify the scope of the analogy as follows:*

*"Water flow along the soil–plant continuum is governed by hydraulic conductivities and water potential gradients. While the water movement in sapwood differs from those in the vadose zone, certain functional similarities exist. For instance, the flow in xylem and soil can both be approximated by Poiseuille's Law (e.g., Nobel, 2009; Jougnot et al., 2019) and exhibit similar responses of hydraulic conductivity to water potential gradients (Cai et al., 2022). Although xylem hydraulic conductivity does not fluctuate with minor variations in soil water potential, significant reductions in soil moisture can lead to declines in xylem conductivity, particularly under drought conditions (Brodribb and Hill, 2000; Carminati and Javaux, 2020). When soil water potential falls within a moderate range, xylem hydraulic conductivity remains relatively stable. However, at lower soil water potentials, xylem conductivity decreases (Kröber et al., 2014; Cai et al., 2022).*
*For the Font-Blanche site, the selected two-week periods shown in Figs. R1-R2 correspond to different seasonal water dynamics (Fig. R7). As indicated in Fig. R7b, soil water potential is lower in spring and summer compared to winter and autumn. Notably, the slopes (ratios) of $VMD_\mathrm{s}^{SP}$ to $VMD_\mathrm{s}^{SV}$ peak during summer, with similar but lower values in winter and autumn. Assuming the xylem conduits behave similarly to porous materials, the empirical relationship between the hydraulic permeability $k$ and $\hat{Q}_v$ for porous materials is expressed by (Jardani et al., 2007):*

$$log_{10}(\hat{Q}_v) = -9.23 - 0.82log_{10}k. \ (12)$$

*Based on Eq. (12) and the $\hat{Q}_v$ estimates introduced in Section 4.3.2, we calculated the sapwood permeabilities for Aleppo pine (FBPh) using Eq. 12 as $4.49\times 10^{-13}$, $2.01\times 10^{-13}$, $1.06\times 10^{-13}$, and $5.38\times 10^{-13}$ m² in winter, spring, summer, and autumn, respectively."*

*In addition, we plan to add a new figure illustrating this relationship, highlighting seasonal variations in the correlation between sap velocity and SP (e.g., Fig. R1). This figure will either be integrated into the main text to reinforce the discussion or included in the Supplementary Materials if space constraints arise.*

[Figure]

*Figure R7: The water surplus and soil matric potential in vicinity of tree through 2023 at the Font-Blanche site. Yellow shaded areas represent the selected periods within different seasons.*

*References:*

*Brodribb, T. J., & Hill, R. S. (2000). Increases in water potential gradient reduce xylem conductivity in whole plants. Evidence from a low-pressure conductivity method. Plant Physiology, 123(3), 1021-1028. https://doi.org/10.1104/pp.123.3.1021*

*Nobel, P. S. (2009). Physicochemical and Environmental Plant Physiology. Academic Press, ISBN 978-0-12-374143-1.*

*Kröber, W., Zhang, S., Ehmig, M., & Bruelheide, H. (2014). Linking xylem hydraulic conductivity and vulnerability to the leaf economics spectrum—a cross-species study of 39 evergreen and deciduous broadleaved subtropical tree species. PLoS One, 9(11), e109211.*

*Jougnot, D., Mendieta, A., Leroy, P., & Maineult, A. (2019). Exploring the effect of the pore size distribution on the streaming potential generation in saturated porous media, insight from pore network simulations. Journal of Geophysical Research: Solid Earth, 124(6), 5315-5335. https://doi.org/10.1029/2018JB017240*

*Carminati, A., & Javaux, M. (2020). Soil rather than xylem vulnerability controls stomatal response to drought. Trends in Plant Science, 25(9), 868-880.*

*Cai, G., Ahmed, M. A., Abdalla, M., & Carminati, A. (2022). Root hydraulic phenotypes impacting water uptake in drying soils. Plant, Cell & Environment, 45(3), 650-663. https://doi.org/10.1111/pce.14259*

L425-430 Xylem and wood is very different from mineral media and made of cellulose, hemi cellulose and lignin. Does any data exist for zeta potential of wood? Isn't the comparison with mineral media is a bit limited by the fact you use eq 12 elaborated for mineral media. As zeta

potential is an important coupling factor at least that points to the need to its experimental estimation.

**[Response]** *We acknowledge the fundamental differences between xylem/wood and mineral media. Xylem is composed primarily of cellulose, hemicellulose, lignin, and pectin, with small amounts of protein and enzymes (e.g., Nobel, 2009; Zhong et al., 2018). In contrast, mineral media typically consist of inorganic materials such as silica. Despite these compositional differences, certain similarities exist in their electrokinetic properties, particularly in surface charge behavior and ion adsorption in the presence of an electrical double layer.*

*For instance, the carboxyl groups in pectin and the plasma membrane are negatively charged, providing cation-binding capacity to the cell walls (Gage et al., 1985, 1986; Hubbe, 2006; Nobel, 2009). However, under specific conditions, xylem cell walls may have few positively charged binding sites (Senden et al., 1992; Kinraide et al., 1998). Similarly, negatively charged mineral surfaces also exhibit high cation-binding capacities and generate negative surface potentials (Revil & Jardani, 2013).*

*Several studies have reported the formation, calculations, and measurements of Zeta potentials in plant systems (e.g., Yermiyahu et al., 1997; Wang et al., 2008, 2011; Kopittke et al., 2014; Lu et al., 2018). For example, Kinraide et al. (1998) collected data showing that Zeta potentials of plant protoplasts and plasma membranes in various media ranges from -48 mV to 23 mV, depending on pH and ionic conditions. This range is smaller but still comparable to Zeta potentials typically observed for silica grains in electrolyte solutions. While direct data on the Zeta potential of xylem is scarce, studies on cellulosic fiber surfaces suggest that their electrokinetic properties similarly exhibit negative zeta potentials under near-neutral pH conditions (Hubbe, 2006).*

*We recognize the limitations of applying equations derived for mineral porous media to biological materials like xylem. However, in the absence of empirical electrokinetic models applied to plant systems, this approach serves as a preliminary step to explore the potential for modeling SP and SV in plants.*

*To address these points, we will revise the manuscript to include the following clarification:*
*"While the electrokinetic properties of plant tissues differ from those of mineral media due to their organic composition, similarities in surface charge behavior and ion adsorption justify the preliminary use of porous media models to estimate SP and SV in plants. Plant tissues, such as xylem cell walls composed of cellulose, hemicellulose, lignin, and pectin, exhibit negatively charged surfaces of plasma membrane and carboxyl groups in pectin, allowing for cation binding. This behavior is comparable to negatively charged mineral surfaces with electrical double layers. Studies have reported Zeta potentials in plant systems ranging from -48 mV to 23 mV, depending on pH and ionic conditions (Kinraide et al., 1998), smaller in magnitude but still comparable to values observed in silica grains. While direct measurements of xylem Zeta potential are limited, research on lignocellulosic materials suggests they also exhibit negative Zeta potentials under near-neutral pH conditions (Hubbe, 2006). Despite these similarities, plant tissues differ fundamentally from mineral media, and their unique composition affects water and ion transport. Experimental studies are essential to directly measure xylem Zeta potential and refine models for plant electrokinetic processes. This would improve the accuracy of SP and SV estimates and advance our understanding of electrokinetic phenomena in vascular plants."*

*References:*

Gage, R. A., Van Wijngaarden, W., Theuvenet, A. P. R., Borst-Pauwels, G. W. F. H., & Verkleij, A. J. (1985). Inhibition of $Rb^+$ uptake in yeast by $Ca^{2+}$ is caused by a reduction in the surface potential and not in the Donnan potential of the cell wall. Biochimica et Biophysica Acta (BBA)-Biomembranes, 812(1), 1-8.

Gage, R. A., Theuvenet, A. R. P., & Borst-Pauwels, G. W. F. H. (1986). Effect of plasmolysis upon monovalent cation uptake, 9-aminoacridine binding and the zeta potential of yeast cells. Biochimica et Biophysica Acta (BBA)-Biomembranes, 854(1), 77-83.

Senden, M. H. M. N., Van Paassen, F. J. M., Van der Meer, A. J. G. M., & Wolterbeek, H. T. (1992). Cadmium—citric acid—xylem cell wall interactions in tomato plants. Plant, Cell & Environment, 15(1), 71-79.

Yermiyahu, U., Rytwo, G., Brauer, D. K., & Kinraide, T. B. (1997). Binding and electrostatic attraction of lanthanum ($La^{3+}$) and aluminum ($Al^{3+}$) to wheat root plasma membranes. The Journal of Membrane Biology, 159, 239-252.

Kinraide, T. B., Yermiyahu, U., & Rytwo, G. (1998). Computation of surface electrical potentials of plant cell membranes: correspondence to published zeta potentials from diverse plant sources. Plant Physiology, 118(2), 505-512.

Hubbe, M. A. (2006). Sensing the electrokinetic potential of cellulosic fiber surfaces. BioResources, 1(1), 116-149.

Wang, P., Zhou, D., Kinraide, T. B., Luo, X., Li, L., Li, D., & Zhang, H. (2008). Cell membrane surface potential ($\psi_0$) plays a dominant role in the phytotoxicity of copper and arsenate. Plant Physiology, 148(4), 2134-2143.

Wang, P., Kinraide, T. B., Zhou, D., Kopittke, P. M., & Peijnenburg, W. J. (2011). Plasma membrane surface potential: dual effects upon ion uptake and toxicity. Plant Physiology, 155(2), 808-820.

Kopittke, P. M., Wang, P., Menzies, N. W., Naidu, R., & Kinraide, T. B. (2014). A web-accessible computer program for calculating electrical potentials and ion activities at cell-membrane surfaces. Plant and Soil, 375, 35-46.

Lu, H. L., Liu, Z. D., Zhou, Q., & Xu, R. K. (2018). Zeta potential of roots determined by the streaming potential method in relation to their Mn (II) sorption in 17 crops. Plant and Soil, 428, 241-251.

L443-450 In Love et al (2008) study the difference in pH is between the soil and the plant, when they show the correlation SP-pH. If a pH for soils of these is considered to be ~8 (calcareous soil) and xylem sap ~6, the SP related to pH would be ~ 120 mV, ie in the order of magnitude of measured SP signal, and electrokinetic effect. Could authors comment on that?

**[Response]** *The measured SP between the xylem sap and the soil results from an integrated bioelectrical process involving both electrochemical and electrokinetic effects along the pathway. Concentration gradients, water potential gradients, and pH differences within the xylem, root, and soil contribute to SP amplitudes. If the pH difference between the measured electrodes is high, the electrochemical effect may dominate SP generation and occupy the major proportion of SP amplitudes. However, the electrokinetic effect related to transpiration would still persist.*

*When measuring SP in sapwood over a small distance (10 cm in our case), the contribution of*

*complex processes is minimized, allowing for a focus on the streaming potential driven by sap flow, as indicated in Fig. 1c of Love et al. (2008). For this short distance, slight pH differences would generate small voltage contributions, estimated at ~ -2.53 mV (Line 447).*

*We introduced the study by Love et al. (2008) in Lines 176–184 of the original manuscript. To address this point further, we will add the following sentence to the Discussion:*

*"If the pH difference between the measured electrodes is significant, the electrochemical effect may dominate SP generation; however, the electrokinetic effect associated with transpiration would still persist."*

L451 metal ions => rather alkaline earth cations

**[Response]** *As suggested by the reviewer, this sentence will be revised to "...including alkaline earth cations ($K^+$, $Ca^{2+}$, $Mg^{2+}$)...".*

L478 481 to be discussed in discussion …

**[Response]** *This sentence will be corrected.*

---

## Author Response (AR1)

**Point-by-point response to the reviews**

*We thank the editor, Anke Hildebrandt, and two anonymous reviewers for their constructive comments that have helped in improving our manuscript.*

*Please find below our responses to each comment. We reproduced all comments here in their entirety and our responses are in BLUE italics. Details concerning what has been changed from our original responses in the open review process are underlined and bold.*

**Comments from the editor**

Dear Damien Jougnot, Kaiyan Hu, and co-authors,

Thank you for submitting this innovative work and well-executed study to HESS. Two reviewers have commented on your manuscript, and both believe that the work is of interest to the HESS readership and of high quality. I concur with their assessment. Reviewer 2 made several constructive comments, the majority of which you have already addressed in your response.

You have made a sincere effort to address the second reviewer's constructive input. However, the public discussion was already closed, so this reviewer could not comment on those changes. I will therefore seek their opinion in a second round. Please implement the changes as suggested and submit a step-by-step response along with a tracked version of the revised manuscript to support a swift review of this revision.

I look forward to receiving your revised manuscript,

Anke Hildebrandt

**[Response]** *We thank the editor for coordinating the thorough peer review process and for forwarding our work to expert reviewers, whose constructive feedback helped in improving our manuscript. In our revised manuscript, we have carefully addressed all the reviewers' comments and incorporated their suggestions to improve the manuscript.*

**Comments from Reviewer #1**

This is an excellent paper form the highly reputed authors.

The manuscript is an easy read, it's logically structured, arguments and thought processes are well founded.

**[Response]** *We thank the reviewer for the comments and positive feedback on our manuscript. Please find our point-by-point response below.*

My only few recommendations would be:

1. Remove reference to the excess charge densities of rocks when discussing this property interpreted for trees. Given the expertise of some authors this is not surprising, but I don't find this comparison relevant. It would be better to compare the data with similar.

   [**Response**] *To discuss the differences and similarities between rocks and trees in electrokinetic processes, we added a paragraph, and some references related to Zeta potentials in plant systems (Lines 503–514).*

2. Add a brief description of the color scheme used in Figs. 4b-c. This could be done as a legend to the colored bar or a brief description in the caption.

   [**Response**] *We thank the referee for this suggestion to enhance the figure's clarity. We have added a label indicating coherence to the colorbar in Figs. 4b–c and provided a brief description in the caption to improve readability. Please refer to the updated Fig. 4 in the revised manuscript.*

[Figure]

*Figure 4: Wavelet coherence analysis between sap velocity and tree SP data at the Font-Blanche site in 2023. (a) Water surplus calculated as the difference between precipitation and actual evapotranspiration. Wavelet coherence maps for the (b) Aleppo pine and (c) Holm oak at the Font-Blanche site with yellow highlighted boxes indicating periods of water surplus. Arrows denote the lag/lead phase between the two timeseries; white dashed lines (b–c) indicate the cone of influence within which edge artifacts are negligible.*

3. Sections 5.2 seems a bit too speculative, so I'd recommend to shorten the discussion and append it to Section 4. However, I will leave the decision on whether to do this or not to the authors.

   **[Response]** *We appreciate the referee's suggestion. **Since previous publications (Love et al., 2008; Gil and Vargas, 2023) have argued that the pH difference dominates the tree SP generation, we discussed the possible electrochemical contribution to our measurements in Section 5.2.** However, since we did not measure pH **of the sap**, this part is **indeed** speculative. As the referee pointed out, Section 5.2 reads more like a hypothesis, so we **shortened** this section **(Lines 529–534)** rather than moving it to Section 4, while ensuring that the Results section remains focused on the **actual** findings derived from **our** data.*

**Comments from Reviewer #2**

**General comments**

This article presents very interesting data and data analysis of a 1 year monitoring of self potential (SP) and sap flow (SF) measurements on 3 different mediterranean forests. 1 or 2 trees are measured in each site, with single SP measurement on each tree. The paper is well written, well organized (except some parts, cf specific comments). It necessitates some preliminary skills in SP, but should be understandable for a broad audience. The strengths of the study are these long term measurements, the pertinent signal analysis used for understanding links (or no links) between SF and SP, which open way for understanding complicated signal from in situ experiments, and the quantifications with theoretical grounds for SP interpretation given by the authors. Exemplarily, the authors shows that it is not so much evident to relate SF and SP despite various studies started in the 60's and show the different processes that can affect the signal. If the strength of the study is in the diversity of site and trees, in my opinion there is a lack of discussion in the interpretation of the results in the discussion according to the different sites and tree species. From experimental point of view, only 1 tree is measured per site (2 at Font Blanche, but 1 tree of each species). If I understand that it could be a high workload, more replicated measured tree sampled would have been interesting, at least, to examine the range of variation for a same location. From another point of view, most of the interpreted results are based on 2 weeks in April for a 1 year study. May be looking at other contrasted periods or seasons would have been interesting for deciphering the different processes acting on SP signal (ie in winter when transpiration is slowed down or halted), in autumn when trees are recovering from possible drought, in summer during the drought and limitation of transpiration by water availability).

I add below some more specific comments, which can also call for some discussion on some points.

**[Response]** *We appreciate the reviewer's positive feedback regarding the data collection, analysis, and overall structure of the manuscript. We understand the concern about the limited number of measured trees per site. The selected trees in this work represent the dominant species at each location. Given the limited knowledge concerning tree SP, we agree with the reviewer that conducting repeated measurements on multiple trees at each site would enhance the understanding of SP characteristics. We are grateful for this suggestion and will implement that for future studies.*

*However, we emphasize that this study presents a unique long-term dataset, integrating SP and sap velocity (SV) measurements __on__ four trees. This dataset offers valuable insights into the relationship between SP and sap flow, as well as the influence of rainfall, over an extended period. By providing open access to this one-year dataset, we aim to encourage further analysis, measurements, and research into tree SP, fostering broader exploration in this relatively understudied area. Our findings highlight the electrokinetic effect as a primary driver of tree SP during dry seasons, demonstrating the method's potential for assessing transpiration rates. We believe this work may lay the foundation for future investigations into tree SP signals. __Considering__ the reviewer's comments, we __expanded Section 5.3 to recommend further SP measurements across multiple trees within the same site to enhance the robustness of the findings (Lines 550–560)__.*

*We also appreciate the reviewer's suggestion to explore SP and SV dynamics during different seasons, particularly in winter, summer, and autumn when __the__ transpiration status are different. While our initial analysis focuses on a two-week period in April—representing active transpiration across different species (Figs. 3c&d)— we acknowledge the importance of seasonal variability in interpreting SP signals. To address this, we have extended our analysis to include data from winter, spring, summer, and autumn, selecting periods without rainfall to avoid confounding effects (Figs. R1–R2).*

*The expanded analysis indicates that SP signals remain responsive throughout the year, albeit with variations in intensity. Increased phase shifts between SV and SP are observed during cooler months (Fig. R1b&h, R2b&h), yet the predicted SP derived from SV data aligns well with decomposed SP observations. Notably, the linear relationship between SV and SP is more pronounced for Aleppo pine compared to Holm oak (Table R1). For both species, SP correlates strongly with sap flow during spring and summer. The slope of the linear model between decomposed SV and SP data peaks in summer (Fig. R1e, R2e), reflecting heightened SP amplitudes despite indications of drought-induced limitations in SV (Figs. R1e&f, R2e&f). Additionally, the raw SV data exhibits an evident decline in July (Fig. 3c–d).*

[Figure]

***Figure R1(Figure 8): The relationship between the fifth decomposed modes of sap velocity ($VMD_5^{SV}$) and tree SP ($VMD_5^{SP}$) signals in (a–b) winter, (c–d) spring, (e–f) summer, and (g–h) autumn of 2023 for the Aleppo pine (FBPh) at the Font-Blanche site. (a, c, e, f) Scatter plots of the decomposed modes of sap velocity and SP signals with black lines indicating linear regression results and red dots indicating points (outliers) outside the 95% confidence level of the regression. (b, d, f, h) Corresponding decompositions of SP signals (orange dots) and calculated SP (black lines) based on the linear relationship between $VMD_5^{SV}$ and $VMD_5^{SP}$, with red dots as outliers outside the 95% confidence level of the linear regression.***

*Furthermore, while this study focuses primarily __on__ electrokinetic effects related to transpiration, we acknowledge the potential influence of other physiological processes. Investigating SP generated by alternative mechanisms, such as ion movement and concentration changes, could yield valuable insights if additional data representing these processes were collected. However, the current dataset primarily includes SV and SP measurements (Fig. 2). As such, we lack direct evidence linking SP to processes beyond transpiration.*

*In response to the reviewer's comment, we **revised** the Discussion section to discuss the seasonal relationships between SP and SV and acknowledge the need for future studies exploring other physiological drivers of SP. **Please see the revised sections 5.1&5.3 of the Discussion part.**

[Figure]

**Figure R2 (Figure S9): The relationship between the fifth decomposed modes of sap velocity ($VMD_5^{SV}$) and tree SP ($VMD_5^{SP}$) signals in (a–b) winter, (c–d) spring, (e–f) summer, and (g–h) autumn of 2023 for the Holm oak (FBQi) at the Font-Blanche site. (a, c, e, f) _Scatter plots_ of the decomposed modes of sap velocity and SP signals with black lines indicating linear regression results and red dots indicating points (outliers) outside the 95% confidence level of the regression. (b, d, f, h) Corresponding decompositions of SP signals (orange dots) and calculated SP (black lines) based on the linear relationship between $VMD_5^{SV}$ and $VMD_5^{SP}$, with red dots as outliers outside the 95% confidence level of the linear regression.**

*Table R1 (Table S2) Seasonal characteristics of two-week data (Feb. 09–23, Apr. 16–30, Jul. 01–15, and Sep. 25–Oct. 09) for the Aleppo pine (FBPh) and Holm oak (FBQi) at the Font-Blanche site.*

| Season | Property | | FBPh | FBQi |
|--------|----------|--|------|------|
| **Winter** | Range | $VMD_5^{SP}$ (mV) | [−4.78, +3.90] | [−21.81, +17.83] |
| | | $VMD_5^{SV}$ (µm·s⁻¹) | [−10.56, +9.86] | [−5.61, +6.49] |
| | Standard Deviation | $VMD_5^{SP}$ (mV) | 1.89 | 7.13 |
| | | $VMD_5^{SV}$ (µm·s⁻¹) | 5.49 | 3.00 |
| | $\hat{Q}_v$ (C·m⁻³) | | 7.9 | 43.2 |
| | Correlation Coefficient | | 0.83 | 0.45 |
| **Spring** | Range | $VMD_5^{SP}$ (mV) | [−5.57, +4.69] | [−19.52, +16.04] |
| | | $VMD_5^{SV}$ (µm·s⁻¹) | [−6.29, +6.36] | [−10.60, +10.61] |
| | Standard Deviation | $VMD_5^{SP}$ (mV) | 2.28 | 7.51 |
| | | $VMD_5^{SV}$ (µm·s⁻¹) | 3.61 | 6.33 |
| | $\hat{Q}_v$ (C·m⁻³) | | 15.2 | 31.2 |
| | Correlation Coefficient | | 0.88 | 0.66 |
| **Summer** | Range | $VMD_5^{SP}$ (mV) | [−5.58, +5.93] | [−26.82, +22.35] |
| | | $VMD_5^{SV}$ (µm·s⁻¹) | [−4.22, +4.09] | [−9.36, +9.32] |
| | Standard Deviation | $VMD_5^{SP}$ (mV) | 2.24 | 11.94 |
| | | $VMD_5^{SV}$ (µm·s⁻¹) | 2.15 | 5.43 |
| | $\hat{Q}_v$ (C·m⁻³) | | 25.7 | 68.0 |
| | Correlation Coefficient | | 0.91 | 0.77 |
| **Autumn** | Range | $VMD_5^{SP}$ (mV) | [−5.94, +5.06] | [−14.73, +11.81] |
| | | $VMD_5^{SV}$ (µm·s⁻¹) | [−10.90, +10.89] | [−4.48, +4.68] |
| | Standard Deviation | $VMD_5^{SP}$ (mV) | 2.44 | 6.68 |
| | | $VMD_5^{SV}$ (µm·s⁻¹) | 6.68 | 2.82 |
| | $\hat{Q}_v$ (C·m⁻³) | | 6.8 | 51.6 |
| | Correlation Coefficient | | 0.69 | 0.54 |

*Below is our detailed point-by-point response to the reviewer's comments.*

**Specific comments**

13 Transpiration is => Plant transpiration is

**[Response]** *We thank the reviewer for this addition. We **revised** the text to specify "Plant transpiration".*

14 Solely relying on sap flow measurements => Not really as at the stand scale (evapo) transpiration can also be assessed by flux towers' measurements.

**[Response]** *We generally agree with the reviewer, but our focus here is on techniques for measuring transpiration at the individual tree level.*

*Flux towers, while effective for estimating ecosystem-scale evapotranspiration, do not directly measure transpiration from individual trees. Instead, flux towers capture the combined water loss from soil evaporation and plant transpiration, typically requiring additional **modelling** or partitioning techniques to isolate transpiration (e.g., Nelson et al., 2020). In contrast, sap flow measurements provide a direct method to quantify transpiration at the individual tree level. To clarify this point, we **revised** this sentence with "**While sap flow measurements offer a direct method for estimating individual tree transpiration, their effectiveness may be limited by the use of point sensors…**" (Lines 14–15).*

*Reference:*

*Nelson, J. A., Pérez-Priego, O., Zhou, S., Poyatos, R., Zhang, Y., Blanken, P. D., ... & Jung, M. (2020). Ecosystem transpiration and evaporation: Insights from three water flux partitioning methods across FLUXNET sites. Global Change Biology, 26(12), 6916-6930.*

32-37 Again cite different methods for (evapo)transpiration measurement: flux towers, soil water balance. With the specifity of sap flow measurement being the direct method for trees for obtaining transpiration.

**[Response]** *We thank the reviewer for this suggestion. In this section, we focus on transpiration measurements at the individual tree level rather than stand-scale evapotranspiration. We **clarified** that there are three primary methods for measuring transpiration **in the revised manuscript (Lines 36–40)**. We **revised** the text to clarify it with "Plant transpiration can be measured using various techniques, including sap flow methods that quantify water movement through the xylem (e.g., Goulden et al., 1994; Granier et al., 1996; Kume et al., 2010), porometry to assess leaf-level transpiration and stomatal conductance (e.g., Zhang et al., 1997; Damour et al., 2010), and flux towers (eddy covariance) that estimate stand-scale evapotranspiration, with tree transpiration inferred by partitioning soil evaporation (e.g., Kurpius et al., 2003; Scanlon and Kustas, 2012)".*

**References**

*Goulden, M. L., & Field, C. B. (1994). Three methods for monitoring the gas exchange of individual tree canopies: ventilated-chamber, sap-flow and Penman-Monteith measurements on evergreen oaks. Functional Ecology, 125-135.*

*Granier, A., Biron, P., Bréda, N., Pontailler, J. Y., & Saugier, B. (1996). Transpiration of trees and forest stands: Short and long-term monitoring using sapflow methods. Global Change Biology, 2(3), 265-274.*

*Zhang, H., Simmonds, L. P., Morison, J. I., & Payne, D. (1997). Estimation of transpiration by single trees: comparison of sap flow measurements with a combination equation. Agricultural and Forest Meteorology, 87(2-3), 155-169.*

*Kurpius, M. R., Panek, J. A., Nikolov, N. T., McKay, M., & Goldstein, A. H. (2003). Partitioning of water flux in a Sierra Nevada ponderosa pine plantation. Agricultural and Forest Meteorology, 117(3-4), 173-192. https://doi.org/10.1016/S0168-1923(03)00062-5*

*Damour, G., Simonneau, T., Cochard, H., & Urban, L. (2010). An overview of models of stomatal conductance at the leaf level. Plant, Cell & Environment, 33(9), 1419-1438. https://doi.org/10.1111/j.1365-3040.2010.02181.x*

*Kume, T., Onozawa, Y., Komatsu, H., Tsuruta, K., Shinohara, Y., Umebayashi, T., & Otsuki, K. (2010). Stand-scale transpiration estimates in a Moso bamboo forest:(I) Applicability of sap flux measurements. Forest Ecology and Management, 260(8), 1287-1294.*

*Scanlon, T. M., & Kustas, W. P. (2012). Partitioning evapotranspiration using an eddy covariance-based technique: Improved assessment of soil moisture and land–atmosphere exchange dynamics. Vadose Zone Journal, 11(3). https://doi.org/10.2136/vzj2012.0025*

42 Gindl et al., 1999;) => suppress last;

**[Response]** *We thank the reviewer for pointing out this typo. The semicolon __is__ removed in the revised version __(Line 50)__.*

46 transpiration processes, which facilitate water and solute transport within the xylem and phloem of trees => Rather : transpiration which relies on the transport of water and solutes in xylem…. Phloem transport is not generated by transpiration but by gradients of concentration of sugars which generate turgor pressure gradients and flow, and can occur without transpiration. So add another sentence, for phloem, decoupled from transpiration

46 , trigger electrokinetic and electro-diffusive effects, => a few words to explain the origin of effects ?

**[Response]** *We thank the reviewer for these two comments. We __reworded__ this part with "__transpiration processes facilitate the transport of water and solutes within the xylem (e.g., Kim et al., 2014). Additionally, sugar concentration gradients can generate turgor pressure differences, driving flow within the phloem (e.g., van Bel, 2003). These natural processes can trigger electrokinetic effects through the advection of net electrical charges and electro-diffusive effects driven by electrochemical potential gradients, leading to the generation of biopotentials and measurable SP signals__" __on Lines 53–57__.*

*Reference:*

*van Bel, A. J. (2003). Transport phloem: low profile, high impact. Plant Physiology, 131(4), 1509-1510. https://doi.org/10.1104/pp.131.4.1509*

*Kim, H. K., Park, J., & Hwang, I. (2014). Investigating water transport through the xylem network in vascular plants. Journal of Experimental Botany, 65(7), 1895-1904. https://doi.org/10.1093/jxb/eru075*

132 low of sap induces a natural electric field in the opposite direction, as depicted in Fig. 1e => no clear indication of opposite electric field in 1e ? Only in the transverse direction of the vessel with the subplot. Could you make the graph clearer for this electric potential difference generation.

**[Response]** *In Fig. 1e, the direction of sap flow is indicated by the upward arrow on the right side, while the downward arrow on the left side represents the direction of the electric field generated by the streaming current. This visualization aligns with the electrical double layer model, where an excess of electric charges exists within the Gouy-Chapman diffuse layer. As sap flow drags these excess positive charges upward, it induces an upward streaming current ($J_{ek}$), resulting in the generation of a downward electric field (Eq. 1). We __added__ a sentence to*

*clarify the electrokinetic mechanism: "For negatively charged cell walls, the streaming current, generated by the upward movement of excess charge within the Gouy–Chapman diffuse layer as sap flows, represents the advective transport of electrical charges, resulting in a net source of current density $J_{ek}$" on Lines 145–147 of the revised manuscript.*

134 no external currents (i.e., electrical flux equals 0). => what means external current here? If there if a flow of ion charges what is the return current so that net current J=0?

**[Response]** *In this context, "external currents" refer to electrical currents generated by sources other than the electrokinetic effect, such as externally injected currents. The phrase "electrical flux equals 0" indicates that the current density produced by the electrokinetic effect is counterbalanced by a conduction current density, maintaining charge conservation and resulting in no net external current.*

*To enhance clarity, we **revised** the sentence to: "This describes a condition where only electrokinetic effects are present, with no contribution from diffusive currents in Eq. (1), ensuring that the net current remains zero" on Lines 150–152.*

137 of the capillary fluid => flowing fluid?

**[Response]** *Yes, it **is** revised to "flowing fluid" (Line 153).*

140 an empirical relationship => for which media?

**[Response]** *We **completed** this sentence to:"...Linde et al., (2007) proposed an empirical relationship between the logarithm of $C_{ek}$ and $\sigma_f$ for porous media" on Line 158.*

147 where k (m2) denotes permeability => for a porous medium?

**[Response]** *We **revised** this sentence to "...k ($m^2$) denotes permeability **of** a porous medium" on Line 163.*

148 under the assumption of 1-D flow, => and neglecting Jdiff in (1)

**[Response]** *As suggested by the reviewer, we **revised** this sentence to "**Under** the assumption of 1–D flow and neglecting $J_{diff}$ in Eq. (1), the current density is solely governed by the electrokinetic effect" on Lines 163–164.*

149- 155 lead to an increase in solute concentration towards the crown => If available, give range of variation in concentration, that should be small as ions are absorbed by cells too..

**[Response]** *The xylem sap typically contains approximately 10 mM of inorganic nutrients, along with organic nitrogen compounds metabolically synthesized in the root (Nobel, 2009). According to McDonald et al. (2002), for a Norway spruce tree in late autumn, the total concentration of amino acids and the minerals magnesium, calcium, and potassium in the xylem sap ranges from 4 mM at the ground level to 6 mM near the crown base. This concentration range aligns with typical salinity levels in pore water (Linde et al., 2007). We **added** the range of variation in sap concentration to the corresponding text (Lines 169&Lines 175-178).*

Table 1 Soil depth is indicated in m (range 0-80 m !), it is rather cm I think … Evapotranspiration seems low, if potential. Is it actual or potential evapotranspiration ?

**[Response]** *We thank the reviewer for pointing out this typo. The unit of soil depth is **centimetre** indeed. The evapotranspiration is "actual evapotranspiration", which **is** clarified in the **revised Table 1**.*

L240 the sap velocity in μm s => the sap velocity is obtained in μm s-1

**[Response]** *This sentence **is** corrected in the revised manuscript (Line 263).*

L243 high-impedance multimeter controlled by a digital data logger => give the reference of manufacturer and model. What is the sampling frequency of recorded voltages?

**[Response]** *The sampling intervals for tree SP measurements are 1 minute at the Larzac and LSBB sites, and 10 minutes at the Font-Blanche site. Detailed measurement parameters, including sampling intervals and units, are provided in Table S1.*

*We also **included** additional information about the equipment used for the measurements. The high-impedance voltmeter, with an input impedance of 20 GΩ, was controlled by a Campbell Scientific CR1000X datalogger **(Lines 265–268)**.*

L244 which length and diameter for screws? Fig 2 it seems it is not screws for wood works. True? Does screws penetrate sapwood or are just in contact with first outside layer of vessels?

**[Response]** *The size of the trunk electrode is shown in Fig. 2f, and the electrodes used are 3 mm diameter screws made of stainless steel **(Line 268)**. As the bark was peeled from the trunk prior to electrode installation (similarly to the installation of the rod of sapflow meters), the screws penetrate the sapwood to a depth of 2 cm beneath the cork cambium (Line **270**).*

L272 In addition to variational mode decomposition, would Fourier spectrum analysis give also similar or complementary results?

**[Response]** *As illustrated in Fig. R3, Fourier spectrum analysis of sap velocity (SV) data reveals similar dominant frequencies to those identified through variational mode decomposition (VMD) (Fig. 5). Both methods indicate the presence of 1-, 2-, 3-, and 4-day$^{-1}$ frequency signals in the SV data.*

*However, as noted in Lines **299–301**, SP measurements are more complex partly because SP is a passive method (while SV relies on actively heating the sap to measure the heat dissipation). They are susceptible to high-frequency electromagnetic noise, temperature fluctuations, trunk wounds, and electrode polarization, which can introduce long-period drifts and complicate the signal composition. Consequently, the SP spectrum (Figs. R3c-d) does not exhibit clear dominant frequencies that can be directly characterized through Fourier analysis.*

*This is the primary reason we employed wavelet transformation and VMD to **analyse** the data, as these methods are more effective in isolating multiple signals from the SP data. To clarify this point, we **revised** the text accordingly **(Lines 301–304)** and **included** Fig. R3 in the Supplementary Material **(Fig. S1)** to explain why VMD was chosen for this study.*

[Figure]

*Figure R3 (Figure S1): Magnitude of sap velocity (SV) and SP spectra obtained using a fast Fourier transform algorithm. (a, c) SV and SP spectra for Aleppo pine (FBPh). (b, d) SV and SP spectra for Holm oak (FBOi). Data were collected from January 1, 2023, to January 1, 2024 at the Font-Blanch site.*

Fig 3 The time axis is labelled as Jan.01, Feb.01.. May be Jan 23, Feb 23… Jan 24 would be better. 30minutes data are shown, daily values would be interting to show (cumulative as proxy of daily flow and average for SP probably)

**[Response]** *We appreciate the recommendation regarding the time axis __labelling__. However, we believe that displaying the first day of each month (e.g., Jan. 01, Feb. 01) provides a clear and consistent representation of the time series. Therefore, we prefer to retain the current time axis __labelling__.*

*In line with your suggestion, we have added the daily average of SP data and the daily cumulative sap flow to Figs. 3c–f to provide a clearer visualization of trends and variations over time.*

[Figure]

*Figure R4 (Figure 3 in the revised manuscript): One-year data collected at the Font-Blanche site at half-hourly intervals from January 1, 2023, to January 1, 2024. (a) Precipitation and air temperature data; (b) Actual evapotranspiration (Actual ET) and vapor pressure deficit (VPD) data; (c–d) Measured sap velocity in blue lines and daily cumulative sap flow in pink lines for the Aleppo pine (FBPh) and the Holm oak (FBQi), respectively; (e–f) Measured SP data in blue lines and daily average of SP data in pink lines for the Aleppo pine (FBPh) and the Holm oak (FBQi), respectively.*

L306 coherence anomalies => Why anomalies ? not just the coherence,   the highest at 1 day, and meaning, I think, that the SP and SV are more linked in terms of variations to each at a 1 day period than other periods.

**[Response]** *We acknowledge that the term "anomalies" may lead to confusion. To improve clarity, we* **revised** *the sentence as follows:*

*"In particular, the one-day period shows high coherence, with arrows pointing left, indicating that SP is negatively correlated with SV at this timescale. This indicates that SP and SV exhibit stronger correlation at a daily timescale compared to other periods"* **(Lines 342–344).**

L307 negative correlation => ?, figure 4 is not correlation. If you refer to linear correlation in S1, put this sentence before to show the difference in data interpretation between wavelet and correlation.

**[Response]** *Wavelet coherence is indeed a localized correlation in time-frequency space, as described for instance by Grinsted et al. (2004). The coherence maps in Figs. 4b–c illustrate the temporal correlation between SV and SP across different time periods. The arrows in Figs. 4b–c represent the phase relationship between SV and SP. Particularly, the rightward arrows (0° phase) indicate in-phase* **behaviour***, where SV and SP vary synchronously. Leftward arrows (180° phase) indicate anti-phase* **behaviour***, signifying negative correlation or out-of-phase variations between SV and SP. Conversely, upward or downward arrows (±90° phase) reflect phase*

*shifts, suggesting that one signal leads or lags the other by a quarter cycle. In particular, the one-day period shows high coherence, with arrows pointing left, indicating that SP is negatively correlated with SV at this timescale. To improve clarity, we **added** a few sentences to Section 4.1.2 **(Lines 336–344)** explaining the phase relationships and the significance of the arrows in Fig. 4.*

*Reference:*

*Grinsted, A., Moore, J. C., & Jevrejeva, S. (2004). Application of the cross wavelet transform and wavelet coherence to geophysical time series. Nonlinear Processes in Geophysics, 11(5/6), 561-566. 1607-7946/npg/2004-11-561*

Fig4 the meaning for arrows as lag phase is not very clear to me. It is not explained and used in the text in 4.1.2. If this data it is needed it should be explained and used.

**[Response]** *As our response to the above comment, we **added** a few sentences **(Lines 338–342)** as follows:*

*"The arrows in Figs. 4b–c represent the phase relationship between SV and SP. Particularly, the rightward arrows (0° phase) indicate in-phase behaviours, where SV and SP vary synchronously. Leftward arrows (180° phase) indicate anti-phase behaviours, signifying negative correlation or out-of-phase variations between SV and SP. Conversely, upward or downward arrows (±90° phase) reflect phase shifts, suggesting that one signal leads or lags the other by a quarter cycle."*

L324 330 Can you comment not only on frequency differences but also on the "power" of signal associated to those frequencies?

**[Response]** *To address the reviewer's comment, we **expanded** the description to discuss not only the frequency differences but also the spectral amplitudes associated with those frequencies.*

*We **revised** the section **on Lines 364–371** as follows:*

*"… noise present in the tree SP data tends to manifest at higher frequencies, as indicated by the prominent spectral amplitudes in the higher-frequency range (Figs. 5a, c, e, g). The low-frequency components, which reflect baseline trends, dominate the last intrinsic modes and occupy the largest proportion of the signals, as shown by the highest amplitudes in the SP and SV spectra (Figs. 5k–l). Apart from the fifth intrinsic mode, the decompositions of SP and SV exhibit distinct frequency characteristics, highlighting the complexity of SP signals. Notably, the fifth intrinsic modes for both datasets reveal a strong diurnal rhythm, with a dominant frequency of $1 \, d^{-1}$ (Figs. 5i–j). The spectra of the fifth decomposition also display the second-highest amplitudes, indicating that these diurnal components represent a significant portion of the overall SP and SV signals"*

Fig 5: x axis could be labelled Frequency (d-1), for better understanding. Indicate what means |u1(f)| … in the legend.

**[Response]** *Done.*

[Figure]

*Figure R5 (Figure 5): Frequency spectra of six decomposed modes of (left column: a, c, e, g, i, k) tree SP and (right column: b, d, f, h, j, l) sap velocity data obtained on the Holm oak (FBQi) at the Font Blanche site within 2023 using the variational method (VMD); Different rows correspond to different modes, where "CF" and "DF" indicate the centre frequency and dominant frequency of the corresponding mode, respectively.*

349 during a two-week period in the growing season => Which 2 weeks and why those 2 weeks?

**[Response]** *The two-week period from April 16 to April 30 was selected to capture a phase of active transpiration across different species, during which no precipitation was recorded (as shown in Fig. 3). We added a sentence on Lines 391–395 to clearly state the rationale for selecting this timeframe.*

L352 – 354 Negative correlation, and flow direction. If OK for the 5th mode of SV, looking at raw data on fig 3, SP of pine is negative, while for oak is positive … (and would lead to a positive and negative correlation), how to reconciliate with your statement?

**[Response]** *The raw SP data for both Aleppo pine and Holm oak indeed show fluctuating trends that span both negative and positive ranges (Figs. 3e–f). This variability arises primarily from long-term electrode drifts and low-frequency influences unrelated to sap flow, as mentioned in Lines 299–300. These drifts can obscure the intrinsic relationship between SV and SP. To address this, we applied wavelet analysis and VMD to isolate SP signals corresponding to the dominant sap flow frequencies. By focusing on the fifth decomposed mode, which shares the same diurnal frequency as sap flow, we effectively removed low-frequency drift and high-frequency noise, allowing us to focus on the intrinsic relationship between SP and SV.*

    *The coherence maps in Fig. 4, where most phase arrows point left, indicate negative correlations between SP and SV in the one-day period. Additionally, the negative ratios of the fifth decomposed modes of SP to SV for both Aleppo pine and Holm oak (Fig. 6a&c) further confirm this negative relationship.*

*We **revised** the manuscript **on Lines 393–395** to include the following clarification: "The large long-term fluctuations in raw SP data across positive and negative ranges, as well as high-frequency interferences, are excluded by extracting the fifth decomposed mode. This allows us to focus on relative variations aligned with the dominant sap flow frequency, providing clearer insight into the relationship between SP and SV."*

L359 and others occurrence in the text : The fifth decomposed signals => rather the fifth decomposed mode of signals. In the text, and fig legend (eg in fig 6) it could be clearer if you use "the fifth decomposed mode of" …

**[Response]** *We are thankful for the reviewer's suggestion. The "the fifth decomposed signals" **is** replaced with "the fifth decomposed mode of". Correspondingly, the caption of Fig. 6 is revised as follow:*

*"**Figure 6: The relationship between the fifth decomposed modes of sap velocity ($VMD_5^{SV}$) and tree SP ($VMD_5^{SP}$) signals in April 16–30, 2023, at the Font-Blanche, LSBB, and Larzac site. (a, c, e, f) Scatter plots of the fifth decomposed modes of sap velocity and SP signals for (a) the Aleppo pine (FBPh), (c) the Holm oak (FBQi) at the Font-Blanche site, (e) the Holm oak at the LSBB (LSQi), and (g) the Pubescent oak at the Larzac (LaQp), with black lines indicating linear regression results and red dots indicating points (outliers) outside the 95% confidence level of the regression. (b, d, f, h) Corresponding decompositions of SP signals (orange dots) and calculated SP (black lines) based on the linear relationship between $VMD_5^{SV}$ and $VMD_5^{SP}$, with red dots as outliers outside the 95% confidence level of the linear regression.**"*

L361 applying linear regression => regression of SP vs SV

**[Response]** *This sentence **is** completed in the revised version **(Line 410)**.*

L360- 364 In Fig 6, the 5$^{th}$ mode of SV shows negative values (almost half of data?), ie negative sap flow. What does that mean from a plant point of view?

**[Response]** *The negative values observed in the fifth decomposed mode of SV (Fig. 6) primarily reflect relative variations around the baseline of signals, rather than indicating actual reverse or negative sap flow. This mode captures diurnal fluctuations and oscillations but does not retain the baseline component, which is represented by the sixth decomposed mode. As a result, negative values in the fifth decomposed mode signify periods of lower sap flow relative to the baseline and not negative flow in a physiological sense.*

*To clarify this point, we **added** the following sentence to the text **(Lines 407–410)**: "Please note that the negative values in the fifth decomposed mode of SV signals represent relative deviations from the baseline, reflecting diurnal fluctuations rather than indicating actual reverse sap flow. The baseline component is captured in the sixth decomposed mode, isolating physical sap flow from the oscillatory variations seen in the fifth decomposed mode."*

Fig 6 in legend: red dots indicating outliers => red dots indicating points (outliers) outside the 95% confidence level of the regression. In the graph => In blue Prec. could b replaced by rain. There is no rains at Font Blanche? Why there is blue dots on graphs (b,d,f…). Indicate r² of correlations (a, c,…)

**[Response]** *To address the comments:*

1. *Rainfall at Font-Blanche:*
   *We confirm that no precipitation occurred at the Font-Blanche site during the analyzed period. To clarify this, we have added the following sentence to Section 4.3.1 **(Lines 391–393)**: "The two-week period from April 16 to April 30 was selected to capture a phase of active transpiration across different species, during which no precipitation was recorded at the Font-Blanche site (as shown in Fig. 3)."*

2. *Labeling of Precipitation (Prec.):*
   *Since precipitation data is presented on the right y-axis in Figs. 6b, d, f, h, we prefer to retain the "Prec." label to clearly denote precipitation values.*

3. *Blue Dots in Graphs (b, d, f, h):*
   *The blue dots represent the edge color of the fifth decomposed mode of SP signals (orange dots). To avoid confusion, we have removed the edge color from the dots, ensuring the focus remains on the primary data points.*

4. *Outliers and Regression:*
   *We revised the caption to:*
   *"Red dots indicate points (outliers) outside the 95% confidence level of the regression."*

5. *R²:*
   *In line with the reviewer's suggestion, we have added R² values to Figs. 6a, c, e, g to provide clearer information about the strength of the correlations.*

*Please see the revised Fig. 6* **_as follows_**:

[Figure]

**_Figure R6 (Figure 6): The relationship between the fifth decomposed modes of sap velocity ($VMD_5^{SV}$) and tree SP ($VMD_5^{SP}$) signals in April 16–30, 2023, at the Font-Blanche, LSBB, and Larzac site. (a, c, e, f) Scatter plots of the fifth decomposed modes of sap velocity and SP signals for (a) the Aleppo pine (FBPh), (c) the Holm oak (FBQi) at the Font-Blanche site, (e) the Holm oak at the LSBB (LSQi), and (g) the Pubescent oak at the Larzac (LaQp), with black lines indicating linear regression results and red dots indicating points (outliers) outside the 95% confidence level of the regression. (b, d, f, h) Corresponding_**

*decompositions of SP signals (orange dots) and calculated SP (black lines) based on the linear relationship between $VMD_5^{SV}$ and $VMD_5^{SP}$, with red dots as outliers outside the 95% confidence level of the linear regression.*

L364 383: All of this text should be put either at the end of the section, or a new result section created such as "excess charge density estimation". Which u is used here? from sap flow or 5th mode of sap flow? Why no calculation for other sites?

**[Response]** *As recommended, we __relocated__ this section to the end of Section 4.3.2 __(Lines 454–466)__ and entitle Section 4.3.2 as "__Relationship between tree SP and sap velocity across four trees__".*

*To clarify the notation, the original **u** refers to the fifth decomposed model of sap velocity signals. To avoid confusion with sap flux or the symbols used in Fig. 5, we __replaced__ u with **v**. Here, $v \approx VMD_5^{SV}$ represents localized (relative) sap velocity derived from the fifth decomposed mode of the SV signals. We __reworded__ this sentence for clarity as follows: "__By converting the localized $\nabla V \approx -\frac{VMD_5^{SP}}{10\,[cm]}$ and $v \approx VMD_5^{SV}$ to the International System of Units, we estimated the effective excess charge density ($\hat{Q}_v = \frac{\sigma \nabla V}{v}\Big|_{J=0}$) using linear fitting slope coefficients $\left(\frac{\nabla V}{v}\right)$__" on Lines 456–457.*

*Regarding the absence of calculations for other sites, the following factors influenced this decision: (1) No resistivity measurements were conducted for Holm oak (LSQi) at LSBB (as noted in Table 1); (2) Both the Larzac and LSBB sites experienced rainfall during the selected periods, resulting in poor correlations between SV and SP (Figs. 6e-h). This indicates that sap flow did not primarily drive SP generation at these locations, limiting the applicability of excess charge density estimation.*

*To clarify this in the manuscript, we __added__ the following sentence __on Lines 457–460__: "__Due to the lack of resistivity measurements at LSBB (LSQi) and the presence of rainfall at the Larzac and LSBB sites during the selected periods, excess charge density calculations were limited to the two trees at the Font-Blanche site, where sap flow consistently dominated SP generation.__"*

L391 – 398 and fig 7 : All of this text is difficult to understand. What is actually done is a difficult to follow, and figure 7 is not well explained…. Could it be more clearly rewritten?

**[Response]** *We __reworded__ the text __(Lines 431–448)__ to improve readability and provide a clearer explanation of the analysis performed as " __Since the amplitude of the calculated SP (derived from the fifth decomposed mode of SV signals, $VMD_5^{SV}$) is lower than that of the fifth decomposed mode of SP signals ($VMD_5^{SP}$) for Holm oak at the Font-Blanche site on April 21–22, and 24 (Fig. 6d), we conducted a test to investigate whether the VMD method reduces the relative amplitude of SV signals during decomposition. To achieve this, we compared the raw SP and SV data to the sum of the diurnal features ($VMD_5^{SP}, VMD_5^{SV}$) and the baseline component ($VMD_6^{SP}, VMD_6^{SV}$) obtained from the VMD-processed data (Figs. 7a–b). The combination of the fifth and sixth decomposed modes is analogous to low-pass filtering, retaining lower-frequency signals. The highest diurnal amplitudes of the raw and processed SV data remain relatively stable, whereas SP data shows notable variations, particularly on April 21–22 and 24.__*

*To further investigate whether the underestimation of SP using $VMD_5^{SV}$ resulted from the exclusion of high-frequency components, we examined the sum of the first to fifth decomposed modes of SV signals, representing a detrending process. In Fig. 7c, the fifth decomposed mode of SP ($VMD_5^{SP}$) was compared with the SP calculated from $\sum_{i=1}^{5} VMD_i^{SV}$. This comparison demonstrated that even when all decomposed modes except the baseline were used to reconstruct diurnal SP from SV data, the higher SP amplitudes observed on April 21–22 and 24 could not be reproduced (Fig. 7c). To corroborate this observation, we repeated the analysis*

*using a finite impulse response (FIR) filter, which yielded similar results (Fig. 7d). This suggests that the elevated SP magnitudes observed on these dates may be influenced by factors other than sap flow."*

*The caption **is** modified to enhance the clarity of this test:*

*Figure 7: Two-week SP data during the 2023 growing season for Holm oak (FBQi) at the Font-Blanche site processed using different approaches. (a) Raw SP data compared with the sum of the last two decomposed models of SP signals; (b) Raw sap velocity data compared with the sum of the last two decomposed models of SV signals; (c) Fifth decomposed mode of SP signals, alongside SP calculated from the sum of the first to fifth decomposed modes of SV signals $(\sum_i^5 VMD_i^{SV})$; (d) SP data filtered using a finite impulse response (FIR) filter (bandpass: $1/48 \ h^{-1}$ to $1/1.5 \ h^{-1}$; filter order: 100) compared with SP calculated from FIR-filtered SV data using the same filter parameters.*

L413-415 analogy between unsaturated soil and plant hydraulic conductivity seems doubtful to me. In soil, variation in water content is linked to filling/emptying of pores? Water content in plants is expressed generally on a (fresh) weight basis, not volumetric. When plant desiccates its volume change but not necessarily saturation. Indeed the plant variation in water content is rather loss of water from cells, with a loss of turgor, and not from xylem which remains full of water except in very dry situations when embolism happens. If it could be the case in dry summer, it might not be that for spring, autumn? At least that should be discussed in such analogy.

**[Response]** *We acknowledge that the analogy between unsaturated soil and plant hydraulic conductivity requires **clarification**. In soils, changes in water content are primarily driven by the filling or draining of pores, leading to variations in saturation. In contrast, the upward water movement in plants is largely confined to the xylem, which typically remains water-filled under normal conditions. Water loss in plants generally occurs at the cellular level, resulting in reduced turgor pressure rather than desaturation of the xylem. Xylem desaturation and embolism occur primarily during extreme drought, which is more common in dry summer conditions. While the dynamics of water movement in sapwood differ from those in the vadose zone, certain functional similarities exist in the flow in xylem and soil could be described by Poiseuille's Law (e.g., Nobel, 2009; Jougnot et al., 2019), and the functional response of hydraulic conductivity to water potential gradients (Cai et al., 2022).*

*To address this, we **revised** the manuscript **(Lines 467–486)** to clarify the scope of the analogy as follows:*

*"Water flow along the soil–plant continuum is governed by hydraulic conductivities and water potential gradients. While the water movement in sapwood differs from those in the vadose zone, certain functional similarities exist. For instance, the flow in xylem and soil can both be approximated by Poiseuille's Law (e.g., Nobel, 2009; Jougnot et al., 2019) and exhibit similar responses of hydraulic conductivity to water potential gradients (Cai et al., 2022). Although xylem hydraulic conductivity does not fluctuate with minor variations in soil water potential, significant reductions in soil moisture can lead to declines in xylem conductivity, particularly under drought conditions (Brodribb and Hill, 2000; Carminati and Javaux, 2020). When soil water potential falls within a moderate range, xylem hydraulic conductivity remains relatively stable. However, at lower soil water potentials, xylem conductivity decreases (Kröber et al., 2014; Cai et al., 2022).*

*For the Font-Blanche site, the selected two-week periods shown in Figs. **8&S9** correspond to different seasonal water dynamics (Fig. **S10**). As indicated in Fig. **S10b**, soil water potential is lower in spring**, and summer compared to winter and autumn. Notably, the slopes (ratios) of $VMD_5^{SP}$ to $VMD_5^{SV}$ peak during summer, with similar but lower values in winter and autumn. **An empirical relationship between the hydraulic permeability k and $\hat{Q}_v$ for porous materials is given by Jardani et al.(2007):***

$log_{10}(\hat{Q}_v) = -9.23 - 0.82 log_{10}k. \ (12)$

***This relationship is also reproduced by a simple physically-based model proposed by Guarracino and Jougnot (2018), which consider a capillary as a first approximation for xylem vessel.*** *It is seen that decreasing permeability leads to an increasing effective excess charge density, subsequently reducing electrical streaming current density (Eq. 3) and weakening the electrokinetic effect. Based on Eq. (12) and the $\hat{Q}_v$ estimates introduced in Section 4.3.2, we calculated the sapwood permeabilities for Aleppo pine (FBPh) as $4.49\times 10^{-13}$, $2.01\times 10^{-13}$, $1.06\times 10^{-13}$ , and $5.38\times 10^{-13}$ m2 in winter, spring, summer, and autumn, respectively.* ***Detailed seasonal characteristics of two-week data for the Aleppo pine (FBPh) and Holm oak (FBQi) at the Font-Blanche site can be found in Table S2.***"

*In addition, we **added** a new figure **in the main text (Fig. 8)** illustrating this relationship, highlighting seasonal variations in the correlation between sap velocity and SP**. Figures. S9 and S10 and Table S2 are** included in the Supplementary Materials.*

[Figure]

***Figure R7 (Figure S10): The water surplus and soil matric potential in vicinity of tree through 2023 at the Font-Blanche site. Yellow shaded areas represent the selected periods within different seasons.***

*References:*

*Brodribb, T. J., & Hill, R. S. (2000). Increases in water potential gradient reduce xylem conductivity in whole plants. Evidence from a low-pressure conductivity method. Plant Physiology, 123(3), 1021-1028. https://doi.org/10.1104/pp.123.3.1021*

*Nobel, P. S. (2009). Physicochemical and Environmental Plant Physiology. Academic Press, ISBN 978-0-12-374143-1.*

*Kröber, W., Zhang, S., Ehmig, M., & Bruelheide, H. (2014). Linking xylem hydraulic conductivity and vulnerability to the leaf economics spectrum—a cross-species study of 39 evergreen and deciduous broadleaved subtropical tree species. PLoS One, 9(11), e109211.*

*Jougnot, D., Mendieta, A., Leroy, P., & Maineult, A. (2019). Exploring the effect of the pore size distribution on the streaming potential generation in saturated porous media, insight from pore network simulations. Journal of Geophysical Research: Solid Earth, 124(6), 5315-5335. https://doi.org/10.1029/2018JB017240*

*Carminati, A., & Javaux, M. (2020). Soil rather than xylem vulnerability controls stomatal response to drought. Trends in Plant Science, 25(9), 868-880.*

*Cai, G., Ahmed, M. A., Abdalla, M., & Carminati, A. (2022). Root hydraulic phenotypes impacting water uptake in drying soils. Plant, Cell & Environment, 45(3), 650-663. https://doi.org/10.1111/pce.14259*

L425-430 Xylem and wood is very different from mineral media and made of cellulose, hemi cellulose and lignin. Does any data exist for zeta potential of wood? Isn't the comparison with mineral media is a bit limited by the fact you use eq 12 elaborated for mineral media. As zeta potential is an important coupling factor at least that points to the need to its experimental estimation.

**[Response]** *We acknowledge the fundamental differences between xylem/wood and mineral media. Xylem is composed primarily of cellulose, hemicellulose, lignin, and pectin, with small amounts of protein and enzymes (e.g., Nobel, 2009; Zhong et al., 2018). In contrast, mineral media typically consist of inorganic materials such as silica. Despite these compositional differences, certain similarities exist in their electrokinetic properties, particularly in surface charge **behaviour** and ion adsorption in the presence of an electrical double layer.*

*For instance, the carboxyl groups in pectin and the plasma membrane are negatively charged, providing cation-binding capacity to the cell walls (Gage et al., 1985, 1986; Hubbe, 2006; Nobel, 2009). However, under specific conditions, xylem cell walls may have few positively charged binding sites (Senden et al., 1992; Kinraide et al., 1998). Similarly, negatively charged mineral surfaces also exhibit high cation-binding capacities and generate negative surface potentials (Revil & Jardani, 2013).*

*Several studies have reported the formation, calculations, and measurements of Zeta potentials in plant systems (e.g., Yermiyahu et al., 1997; Wang et al., 2008, 2011; Kopittke et al., 2014; Lu et al., 2018). For example, Kinraide et al. (1998) collected data showing that Zeta potentials of plant protoplasts and plasma membranes in various media ranges from −48 mV to 23 mV, depending on pH and ionic conditions. This range is smaller but still comparable to Zeta potentials typically observed for silica grains in electrolyte solutions. While direct data on the Zeta potential of xylem is scarce, studies on cellulosic fiber surfaces suggest that their electrokinetic properties similarly exhibit negative zeta potentials under near-neutral pH conditions (Hubbe, 2006).*

*We recognize the limitations of applying equations derived **from** mineral porous media to biological materials like xylem. However, in the absence of empirical electrokinetic models applied to plant systems, this approach serves as a preliminary step to explore the potential for modeling SP and SV in plants.*

*To address these points, we **revised** the manuscript to include the following clarification **(Lines 503–514):** "While the electrokinetic properties of plant tissues differ from those of mineral media due to their organic composition, similarities in surface charge **behaviour** and ion adsorption justify the preliminary use of porous media models to estimate SP and SV in plants. Plant tissues, such as xylem cell walls composed of cellulose, hemicellulose, lignin, and pectin, exhibit negatively charged surfaces of plasma membrane and carboxyl groups in pectin, allowing for cation binding. This **behaviour** is comparable to negatively charged mineral surfaces with electrical double layers. Studies have reported Zeta potentials in plant systems ranging from −48 mV to 23*

*mV, depending on pH and ionic conditions (Kinraide et al., 1998**), which, while smaller in magnitude, remain comparable to those** observed in silica grains. While direct measurements of xylem Zeta potential are limited, research on lignocellulosic materials suggests they also exhibit negative Zeta potentials under near-neutral pH conditions (Hubbe, 2006). Despite these similarities, plant tissues **fundamentally** differ from mineral media, **with** their unique composition **influencing** water and ion transport. Experimental studies are essential to directly measure xylem Zeta potential and refine models for plant electrokinetic processes. This would improve the accuracy of SP and SV estimates and advance our understanding of electrokinetic phenomena in vascular plants."*

*References:*

*Gage, R. A., Van Wijngaarden, W., Theuvenet, A. P. R., Borst-Pauwels, G. W. F. H., & Verkleij, A. J. (1985). Inhibition of $Rb^+$ uptake in yeast by $Ca^{2+}$ is caused by a reduction in the surface potential and not in the Donnan potential of the cell wall. Biochimica et Biophysica Acta (BBA)-Biomembranes, 812(1), 1-8.*

*Gage, R. A., Theuvenet, A. R. P., & Borst-Pauwels, G. W. F. H. (1986). Effect of plasmolysis upon monovalent cation uptake, 9-aminoacridine binding and the zeta potential of yeast cells. Biochimica et Biophysica Acta (BBA)-Biomembranes, 854(1), 77-83.*

*Senden, M. H. M. N., Van Paassen, F. J. M., Van der Meer, A. J. G. M., & Wolterbeek, H. T. (1992). Cadmium—citric acid—xylem cell wall interactions in tomato plants. Plant, Cell & Environment, 15(1), 71-79.*

*Yermiyahu, U., Rytwo, G., Brauer, D. K., & Kinraide, T. B. (1997). Binding and electrostatic attraction of lanthanum ($La^{3+}$) and aluminum ($Al^{3+}$) to wheat root plasma membranes. The Journal of Membrane Biology, 159, 239-252.*

*Kinraide, T. B., Yermiyahu, U., & Rytwo, G. (1998). Computation of surface electrical potentials of plant cell membranes: correspondence to published zeta potentials from diverse plant sources. Plant Physiology, 118(2), 505-512.*

*Hubbe, M. A. (2006). Sensing the electrokinetic potential of cellulosic fiber surfaces. BioResources, 1(1), 116-149.*

*Wang, P., Zhou, D., Kinraide, T. B., Luo, X., Li, L., Li, D., & Zhang, H. (2008). Cell membrane surface potential ($\psi_0$) plays a dominant role in the phytotoxicity of copper and arsenate. Plant Physiology, 148(4), 2134-2143.*

*Wang, P., Kinraide, T. B., Zhou, D., Kopittke, P. M., & Peijnenburg, W. J. (2011). Plasma membrane surface potential: dual effects upon ion uptake and toxicity. Plant Physiology, 155(2), 808-820.*

*Kopittke, P. M., Wang, P., Menzies, N. W., Naidu, R., & Kinraide, T. B. (2014). A web-accessible computer program for calculating electrical potentials and ion activities at cell-membrane surfaces. Plant and Soil, 375, 35-46.*

*Lu, H. L., Liu, Z. D., Zhou, Q., & Xu, R. K. (2018). Zeta potential of roots determined by the streaming potential method in relation to their Mn (II) sorption in 17 crops. Plant and Soil, 428, 241-251.*

L443-450 In Love et al (2008) study the difference in pH is between the soil and the plant, when they show the correlation SP-pH. If a pH for soils of these is considered to be ~8 (calcareous soil) and xylem sap ~6, the SP related to pH would be ~ 120 mV, ie in the order of magnitude of measured SP signal, and electrokinetic effect. Could authors comment on that?

**[Response]** *The measured SP between the xylem sap and the soil results from an integrated bioelectrical process involving both electrochemical and electrokinetic effects along the pathway. Concentration gradients, water*

*potential gradients, and pH differences within the xylem, root, and soil contribute to SP amplitudes. If the pH difference between the measured electrodes is high, the electrochemical effect may dominate SP generation and occupy the major proportion of SP amplitudes. However, the electrokinetic effect related to transpiration would still persist.*

*When measuring SP in sapwood over a small distance (10 cm in our case), the contribution of complex processes is minimized, allowing for a focus on the streaming potential driven by sap flow, as indicated in Fig. 1c of Love et al. (2008). For this short distance, slight pH differences would generate small voltage contributions, estimated at ~ −2.53 mV (__Line 531__).*

*We introduced the study by Love et al. (2008) in Lines __199–201__ of the original manuscript. To address this point further, we __added__ the following sentence to the Discussion __(Lines 527–528)__:*

__"…the electrochemical effect could dominate SP generation under significant pH contrasts between the measured electrodes; however, the electrokinetic effect associated with transpiration would still persist"__

L451 metal ions => rather alkaline earth cations

**[Response]** *As suggested by the reviewer, this sentence __is__ revised to "...__upward longitudinal concentration gradients of__ alkaline earth cations ($K^+$, $Ca^{2+}$, $Mg^{2+}$)..."__(Lines 532–535)__.*

L478 481 to be discussed in discussion …

**[Response]** __We reworded these sentences in the revised Conclusion section (Lines 562–567).__